# PlanMoGPT: Flow-Enhanced Progressive Planning for Text to Motion Synthesis

## Abstract

Recent advances in large language models (LLMs) have enabled breakthroughs in many multimodal generation tasks, but a significant performance gap still exists in text-to-motion generation, where LLM-based methods lag far behind non-LLM methods. We identify the granularity of motion tokenization as a critical bottleneck: fine-grained tokenization induces local dependency issues, where LLMs overemphasize short-term coherence at the expense of global semantic alignment, while coarse-grained tokenization sacrifices motion details. To resolve this issue, we propose PlanMoGPT, an LLM-based framework integrating progressive planning and flow-enhanced fine-grained motion tokenization. First, our progressive planning mechanism leverages LLMs' autoregressive capabilities to hierarchically generate motion tokens by starting from sparse global plans and iteratively refining them into full sequences. Second, our flow-enhanced tokenizer doubles the downsampling resolution and expands the codebook size by eight times, minimizing detail loss during discretization, while a flow-enhanced decoder recovers motion nuances. Extensive experiments on text-to-motion benchmarks demonstrate that PlanMoGPT achieves state-of-the-art performance, improving FID scores by 63.8% (from 0.380 to 0.141) on long-sequence generation while enhancing motion diversity by 49.9% compared to existing methods. The proposed framework successfully resolves the diversity-quality trade-off that plagues current non-LLM approaches, establishing new standards for text-to-motion generation.

## 1 Introduction

Large language models (LLMs) have demonstrated remarkable capabilities across diverse tasks Brown et al. (2020), including multimodal scenarios Liang et al. (2024); Achiam et al. (2023); Alayrac et al. (2022); Liu et al. (2023), by leveraging world knowledge and reasoning abilities acquired through large-scale pretraining Wei et al. (2022); Kojima et al. (2022). Their capacity for autoregressive planning and commonsense reasoning has proven critical for complex multimodal generation tasks Zhao et al. (2023); Wang et al. (2024a), such as image generation Koh et al. (2023) and speech synthesis Zhang et al. (2023a). However, in text-to-motion generation, LLM-based methods still underperform compared to non-LLM approaches (e.g., diffusion-based methods Tevet et al. (2023); Zhang et al. (2022; 2023c); Chen et al. (2023); Shafir et al. (2024)), although the latter still faces limitations such as slow inference speed and low diversity. This paradox motivates our investigation into a fundamental research question: What limits the potential of LLMs in motion generation? Addressing this bottleneck may overcome the limitations of current text-to-motion generation approaches while establishing a new paradigm for efficient, high-quality motion generation.

We identify the granularity of motion tokenization as a core challenge. Specifically, previous LLM-based motion generation approaches typically follow the token-based paradigm. In this paradigm, it first uses a motion tokenizer (e.g., VQ-VAE Van Den Oord et al. (2017)) to compress motion sequences into discrete tokens. LLMs are then trained to generate these tokens, and finally, a decoder is used to reconstruct the motion from generated tokens. Although this approach can incorporate continuous motion into the LLM generation paradigm, motion is represented as a sequence of fine-grained tokens, and similar motions are divided into similar tokens. This causes adjacent tokens to be very similar, which makes the newly generated token provide a strong contextual signal for the next token prediction, thus letting the model ignore the text and earlier tokens, leading to a local dependency problem. As evidenced in Figure 1, for a complex and long motion description,

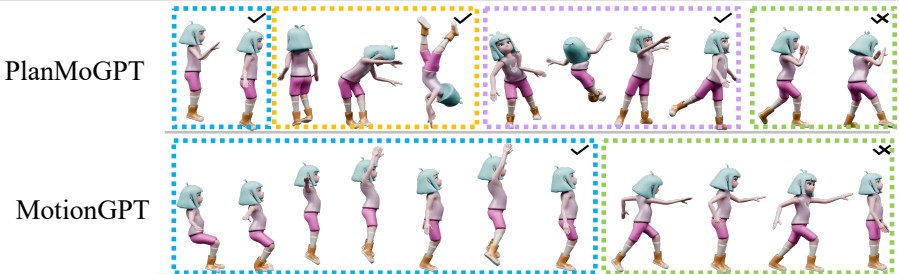

*Textual description: A person began jumping rope, then spun around twice and did a handstand. After that, this person somersaults forward and balances on one leg, then throws a punch and defenses a kick.*

Figure 1: Generating complex and long-sequence motion by our PlanMoGPT, MotionGPT Jiang et al. (2023). The bounding box of each motion clip is color-coded to match its corresponding text.

MotionGPT Jiang et al. (2023) frequently exhibits partial action generation (too much jumping but omitting other actions), as it overemphasizes local motion coherence at the expense of global alignment. Although coarse-grained tokenization can alleviate the local dependency problem, this will lead to the loss of motion details Zhang et al. (2023b) and even make the generated motion discontinuous. This shows that there is a conflict between preserving motion details in motion discretization and generating token sequences stably by LLMs.

To address the above challenges, we propose an LLM-based motion generation framework named PlanMoGPT, which integrates an LLM-based progressive planning mechanism with flow-enhanced fine-grained motion tokenization. First, we resolve local dependency limitations in fine-grained generation through the proposed progressive planning mechanism. Leveraging LLMs' planning capabilities, the system first generates motion tokens at large intervals (e.g., every 4 frames) to establish global motion structure, as a preliminary plan. This plan is then progressively refined to middle-grained (2-frame intervals) and until full-sequence through iterative prediction. Our experimental results show that this progressive planning enables the LLM to generate high-fidelity fine-grained token sequences, even when handling long-sequence motion generation tasks. Building on this foundation, we develop a flow-enhanced fine-grained motion tokenizer with minimal detail loss. Firstly, we employ the VQ-VAE to encode the motion, increasing the codebook size to eight times while reducing the downsampling rate by half compared to prior works Zhang et al. (2023b); Jiang et al. (2023). During motion decoding, we employ a flow-enhanced motion decoder to establish gradient-based mapping from coarse to refined motions, effectively recovering motion details lost during tokenization. Experimental results indicate that our proposed PlanMoGPT significantly outperforms all LLM-based and non-LLM baselines in all metrics, including generation quality, semantic alignment, and diversity, on multiple text-to-motion datasets. This multi-interval LLM-based planning approach also balances high diversity with good semantic alignment simultaneously.

Our contributions are threefold: (1) We propose a text-to-motion generation framework named Plan-MoGPT, which better leverages LLMs by addressing the challenge of the granularity of motion tokenization; (2) We propose a progressive planning method to address the local dependency problem in LLMs, and a flow-enhanced fine-grained motion tokenizer to encode finer-grained motions; (3) Experimental results indicate that PlanMoGPT achieves state-of-the-art performance across multiple text-to-motion benchmarks. Particularly significant improvements are observed on long-sequence datasets, where FID improves from 0.380 to 0.141. Experimental results show that PlanMoGPT resolves the diversity-quality dilemma in existing non-LLM approaches. It generates highly diverse motions, with a 49.9% improvement in diversity metrics over the state-of-the-art baselines, while maintaining superior generation quality in FID scores.

## 2 RELATED WORK

### 2.1 HUMAN MOTION SYNTHESIS

Human motion synthesis aims to generate semantic-aligned and realistic motions conditioned on multimodal inputs, including texts Zhu et al. (2023); Guo et al. (2022); Hong et al. (2022); Guo et al. (2024); Zhang et al. (2023d; 2022; 2023c); Pinyoanuntapong et al. (2024), action labels Yu et al. (2020); Degardin et al. (2022); Guo et al. (2020); Petrovich et al. (2021), scene contexts Cao

et al. (2020); Wang et al. (2021); Hassan et al. (2021); Taheri et al. (2022); Wu et al. (2022), speech signals Ao et al. (2023); Yang et al. (2023); Kucherenko et al. (2019); Li et al. (2021a); Liu et al. (2022), and partial motion sequences Li et al. (2021b); Punnakkal et al. (2021); Ginosar et al. (2019); Chang et al. (2017). Recent text-to-motion approaches primarily employ four generative paradigms: GANs Goodfellow et al. (2014); Xu et al. (2023); Men et al. (2022), VAEs Kingma (2013); Van Den Oord et al. (2017), diffusion models Ho et al. (2020); Weng (2021); Song et al. (2020), and token-based Zhang et al. (2023b); Guo et al. (2024); Pinyoanuntapong et al. (2024); Wu et al. (2025), with diffusion and token-based methods emerging as dominant trends. To align generated motions with texts well, ReMoDiffuse Zhang et al. (2023c) guides the generation through retrieved reference motion, but this may introduce retrieval model bias. FineMoGen Zhang et al. (2023d) introduces time-aligned text decomposition but may compromise global motion coherence. However, the iterative generation in diffusion methods leads to inefficient inference. Recent acceleration attempts use Flow Matching hu2 (2023); Lipman et al. (2022) for faster inference but sacrifice motion quality. Compared with previous works, we address the granularity of motion tokens. We propose the LLM-based progressive planning with a fine-grained flow-enhanced motion tokenizer, which not only delivers state-of-the-art generation quality but also demonstrates substantially faster inference speeds compared to mainstream diffusion models.

### 2.2 Motion-aware Large Language Models

Motion-aware Large Language Models (LLMs) represent the approach that leverages large language models (LLMs) to enhance the understanding and generation of human motion Jiang et al. (2023); Wang et al. (2024b); Wu et al. (2025); Ouyang et al. (2025); Jiang et al. (2024). Inspired by the success of multimodal models Radford et al. (2021); Jia et al. (2021), frameworks such as MotionLLM Wu et al. (2025) and MotionGPT Wang et al. (2024b); Jiang et al. (2023) propose a unified architecture that encodes 3D actions into discrete tokens, enabling LLMs to generate and interpret complex motion sequences effectively. Think-Then-React Tan et al. (2025) leverages LLMs for inferring action intentions and reasoning about reaction descriptions, thereby improving bidirectional human interaction. Motion-R1 Ouyang et al. (2025) incorporates reasoning and reinforcement learning methods to enhance motion generation, adapting the Group Relative Policy Optimization(GRPO) Guo et al. (2025) algorithm to the domain of motion synthesis. Additionally, MotionChain Jiang et al. (2024) integrates multi-turn conversational capabilities to control continuous virtual human movements, facilitating more natural and coherent interactions. Despite these advancements, challenges remain in achieving robust long-term action coherence and diversity, especially in complex dynamic scenarios.

## 3 Method

Given a textual description $x$, our goal is to generate a high-quality and semantically aligned motion $y = \{p_i\}_{i=1}^n$, where each pose $p_i \in \mathbb{R}^{d_m}$ is a $d_m$-dimensional vector capturing the position information of human joints at the i-th frame. As shown in Figure 2, our PlanMoGPT consists of two parts: Flow-enhanced motion tokenizer converts motion into token sequence with minimal detail loss (Sec 3.1), and an LLM integrate with progressive planning for token sequence generation (Sec 3.2).

### 3.1 Flow-Enhanced Motion Tokenizer

As the prerequisite stage of our framework, the tokenizer must first establish an accurate discrete motion representation for the following LLM-based motion generation. We first introduce the baseline approach and its limitations, and then introduce the motion tokenizer we use, including fine-grained motion encoding and flow-enhanced motion decoding.

**Baseline Architecture.** Previous approaches Jiang et al. (2023) process input motion $y \in \mathbb{R}^{n \times d_m}$ through a CNN-based encoder with stride $r = 4$, producing latent vectors $Z = \{z_i\}_{i=1}^l$, where $l = \lfloor n/r \rfloor$. These are quantized via nearest-neighbor lookup in a codebook $\mathcal{C} \in \mathbb{R}^{K \times d}$ (typically $K = 512$), generating discrete token sequence $M = \{m_i\}_{i=1}^l$. Then, through a CNN-based decoder, $M$ is reconstructed into motion sequence, which can be denoted as $y_0 \in \mathbb{R}^{n \times d_m}$. While effective for representation, this paradigm suffers from two limitations: 1) High downsampling rates discard temporal resolution, and 2) Small codebooks restrict expressiveness, causing loss of motion details during quantization. Although previous works have also attempted to use more fine-grained tokens,

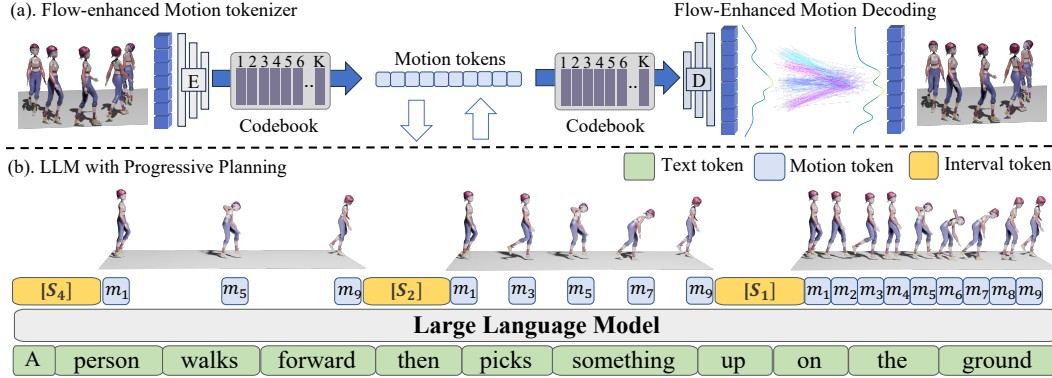

Figure 2: PlanMoGPT consists of two components: (a). A flow-enhanced motion tokenizer converts motion into fine-grained tokens with minimal loss; (b). An LLM integrates with progressive planning, which progressively generates from a larger interval motion tokens to the full motion token sequence.

such as T2M-GPT Zhang et al. (2023b) using a larger codebook, this even causes generated motion quality deterioration, due to downstream generation models lacking sufficient capacity.

**Fine-Grained Motion Encoding.** We improve the previous VQ-VAE through two improvements. First, we reduce the downsampling rate to $r = 2$. This preserves a two-times temporal resolution compared to conventional approaches. To better capture motion details, we incorporate additional convolutional layers with a stride of 1, maintaining fidelity throughout the encoding process. Secondly, we enlarge the codebook size to 4096, eight times larger than previous works. The enlarged code space can represent more motion details, thereby reducing the quantization loss.

**Flow-Enhanced Motion Decoding.** While the motion encoding enables fine-grained motion tokenization, its CNN-based decoder's output $y_0$ still exhibits details losses due to quantization. To bridge this quality gap, we propose to refine $y_0$ through flow matching Lipman et al. (2022) – a continuous optimization process that progressively adds details to the coarse "motion sketch." Specifically, as shown in Figure 2, we leverage the flow-matching method to establish a smooth optimization trajectory that evolves the $y_0$ into the real motion sequence $y$ (now denoted as $y_1$). Guided by a learnable vector field $\mathbf{F}_\theta$, this process is formalized as an ordinary differential equation (ODE):

$$\frac{dy_t}{dt} = \mathbf{F}_\theta(y_t, t), \quad t \in [0, 1] \tag{1}$$

Here, $t$ serves as a virtual time variable: $y_t = y_0$ (coarse motion) at $t = 0$, and $y_t = y_1$ (real motion) at $t = 1$. The vector field $\mathbf{F}_\theta$ acts as a correction signal generator, predicting both the direction and magnitude of adjustments needed for the current motion sequence $y_t$. We optimize $\mathbf{F}_\theta$ using conditional flow matching:

$$\mathcal{L}_{\text{CFM}}(\theta) = \mathbb{E}_{t, y_1, y_t} \left\| \mathbf{F}_\theta(y_t, t) - u(y_t | y_1) \right\|^2, \tag{2}$$

where $u(y_t | y_1)$ represents the tangent direction of the shortest path between $y_t$ and $y_1$. This loss function forces the network to learn the most efficient incremental corrections.

In inference, we discretize the ODE into $T$ refinement steps. Starting from $y_0$, we iteratively update:

$$y_{t_{i+1}} = y_{t_i} + \mathbf{F}_\theta(y_{t_i}, t_i) \cdot \Delta t, 0 = t_0 < \cdots < t_T = 1 \tag{3}$$

where $\Delta t$ is the time increment. This iterative scheme progressively injects motion details (e.g., natural limb oscillations) into the motion sequence $y_0$.

## 3.2 LLM with Progressive Planning

Building on our flow-enhanced motion tokenizer, we design an LLM-based progressive planning mechanism to address the local dependency problem in fine-grained token generation. Our approach consists of two components: Multi-interval plan sampling and a progressive generation pipeline.

| Dataset | Motions | Texts | Texts per Motion | Frame Rate (FPS) | Avg. Duration (s) | Max Duration (s) | Vocab Size |
|---|---|---|---|---|---|---|---|
| KIT-ML | 3,911 | 6,278 | 1–4 | 12.5 | 6.1 | 10 | 1,623 |
| KIT-ML++ | 24,234 | 25,833 | 1–4 | 12.5 | 27.3 | 50 | 2,297 |
| HumanML3D | 14,616 | 44,970 | 3 | 20 | 7.1 | 10 | 5,317 |
| HumanML3D++ | 127,990 | 143,728 | 1–3 | 20 | 20.1 | 50 | 8,593 |

Table 1: Statistics of the text-to-motion generation datasets.

**Multi-Interval Plan Sampling.** Given a motion token sequence $M = \{m_i\}_{i=1}^{l}$, we first construct multi-granularity motion plans through interval sampling:

$$M_{b;T} = \{m_{b+kT} | 0 \leq k \leq \lfloor (l-b)/T \rfloor\}, \tag{4}$$

where $T \in \{4, 2, 1\}$ controls temporal granularity (larger $T$ indicates coarser plans). The base offset $b$ is randomly sampled from $\{1, .., T_{\max}\}$ to ensure diverse plan initialization. This creates a three-level hierarchy: 4-frame interval plans for global structure, 2-frame plans for intermediate refinement, and full-sequence tokens for detail synthesis.

**Progressive Generation Pipeline.** We reformulate motion generation as a hierarchical sequence prediction task by combining multi-interval plans as follows:

$$U = [S_4] \oplus M_{b;4} \oplus [S_2] \oplus M_{b;2} \oplus [S_1] \oplus M, \tag{5}$$

where $\oplus$ denotes concatenation and $[S_i]$ are special tokens marking granularity transitions.

This structure enables the following advantages:

1) **Coarse-to-Fine Generation**: The 4-frame plans ($T = 4$) force the LLM to focus on global semantics by removing 75% of surrounding tokens, establishing text-aligned motion skeletons. Then the 2-frame plans ($T = 2$) inject mid-scale kinematics while maintaining semantic consistency through cross-attention with coarser plans, and finally, the full-sequence generation ($T = 1$) focuses on detail completion under dual constraints from both upper levels;

2) **Cross-Level Error Correction**: The generation of full sequence is conditioned on both upper-level plans, with cross-attention weights showing 32.9% and 15.3% allocation to $M_{b;4}$ and $M_{b;2}$. By conditioning the generation on the continued guidance from the plans, cumulative errors in fine-grained generation can be rectified through backward checks with coarser plans.

Overall, this progressive planning mechanism transforms motion generation from conventional token-by-token prediction to structured plan evolution. As shown in our experiments, the hierarchy-aware generation process achieves better long-term consistency than standard autoregressive approaches while maintaining the detail-preserving capability of fine-grained tokenization.

## 4 EXPERIMENTS

### 4.1 EXPERIMENTAL SETUP

**Datasets.** We conduct evaluations on four datasets, with the statistics shown in Table 1. **HumanML3D** Guo et al. (2022) is formed by text annotation of the HumanAct12 Guo et al. (2020) and AMASS Mahmood et al. (2019) datasets, containing 14,616 motions and 44,970 text descriptions at 20 FPS with an average duration of 7.1 seconds. **KIT-ML** Mandery et al. (2015) is an early dataset (3,911 motions, 6,278 texts) captured at 12.5 FPS from the sources of KIT and CMU Carnegie Mellon University (2003). To further verify the ability to generate complex and long motions, we introduce **HumanML3D++** and **KIT-ML++**, two larger-scale long motions datasets. We follow previous work Li et al. (2024) to merge 2-5 motions to long motions, generating seamless sequences of up to 50s duration (5 times the original maximum). We use GPT-4 to merge texts. We preserve the original data splits to ensure fair benchmarking. We sample 100 motion-text pairs, of which 86% of the sequences are reliable, evaluating both motion smoothness and text-motion alignment. The newly collected datasets achieve an expansion of 8 times (127K+ motions for HumanML3D++, 24K+ motions for KIT-ML++) with an average duration of 20.1s (2.8 times) and 27.3s (4.5 times).

**Implementation Details.** We develop the flow-enhanced tokenizer based on previous approaches Guo et al. (2024); Lipman et al. (2022), with a learning rate of 2e-4, a batch size of 256, and up to 50

| Datasets | Methods | R-Precision ↑ | | | FID ↓ | MM-Dist ↓ | MModality ↑ |
|---|---|---|---|---|---|---|---|
| | | Top-1 | Top-2 | Top-3 | | | |
| Human ML3D | MDM§ | - | - | $0.611^{\pm.007}$ | $0.544^{\pm.044}$ | $5.566^{\pm.027}$ | $2.799^{\pm.072}$ |
| | MFM§ | - | - | $0.642^{\pm.003}$ | $0.362^{\pm.006}$ | $5.280^{\pm.009}$ | $2.443^{\pm.070}$ |
| | MotionDiffuse§ | $0.491^{\pm.001}$ | $0.681^{\pm.001}$ | $0.782^{\pm.001}$ | $0.630^{\pm.001}$ | $3.113^{\pm.001}$ | $1.553^{\pm.042}$ |
| | ReMoDiffuse§ | $0.510^{\pm.005}$ | $0.698^{\pm.006}$ | $0.795^{\pm.004}$ | $0.103^{\pm.004}$ | $2.974^{\pm.016}$ | $1.795^{\pm.043}$ |
| | T2M-GPT | $0.491^{\pm.003}$ | $0.680^{\pm.003}$ | $0.775^{\pm.002}$ | $0.116^{\pm.004}$ | $3.118^{\pm.011}$ | $1.856^{\pm.011}$ |
| | T2M-GPT* | $0.494^{\pm.003}$ | $0.684^{\pm.002}$ | $0.777^{\pm.003}$ | $0.130^{\pm.004}$ | $3.109^{\pm.006}$ | $2.363^{\pm.651}$ |
| | MotionGPT | $0.492^{\pm.003}$ | $0.681^{\pm.003}$ | $0.778^{\pm.002}$ | $0.232^{\pm.008}$ | $3.096^{\pm.008}$ | $2.008^{\pm.084}$ |
| | MotionLLM | $0.515^{\pm.004}$ | - | $0.801^{\pm.004}$ | $0.230^{\pm.009}$ | $2.967^{\pm.020}$ | - |
| | MotionChain | $0.504^{\pm.003}$ | $0.617^{\pm.002}$ | $0.790^{\pm.003}$ | $0.248^{\pm.009}$ | $3.033^{\pm.010}$ | $1.727^{\pm.014}$ |
| | MoMask (base) § | $0.504^{\pm.004}$ | $0.699^{\pm.006}$ | $0.797^{\pm.004}$ | $0.082^{\pm.008}$ | $3.050^{\pm.013}$ | $1.050^{\pm.061}$ |
| | MoMask§ | $0.521^{\pm.002}$ | $0.713^{\pm.002}$ | $0.807^{\pm.002}$ | $\mathbf{0.045}^{\pm.002}$ | $2.958^{\pm.008}$ | $1.241^{\pm.040}$ |
| | BAMM (base) § | $0.507^{\pm.003}$ | $0.703^{\pm.002}$ | $0.799^{\pm.002}$ | $0.111^{\pm.005}$ | $3.028^{\pm.009}$ | $1.750^{\pm.057}$ |
| | BAMM§ | $\underline{0.522}^{\pm.002}$ | $\underline{0.715}^{\pm.002}$ | $\underline{0.808}^{\pm.002}$ | $0.055^{\pm.002}$ | $\underline{2.936}^{\pm.008}$ | $1.732^{\pm.055}$ |
| | PlanMoGPT (base) | $0.521^{\pm.003}$ | $0.711^{\pm.002}$ | $0.804^{\pm.002}$ | $0.106^{\pm.004}$ | $2.937^{\pm.008}$ | $\underline{2.814}^{\pm.235}$ |
| | **PlanMoGPT** | $\mathbf{0.526}^{\pm.002}$ | $\mathbf{0.716}^{\pm.002}$ | $\mathbf{0.809}^{\pm.002}$ | $\underline{0.048}^{\pm.002}$ | $\mathbf{2.884}^{\pm.007}$ | $\mathbf{2.971}^{\pm.227}$ |
| Human ML3D++ | T2M-GPT | $0.357^{\pm.002}$ | $0.509^{\pm.001}$ | $0.590^{\pm.001}$ | $0.651^{\pm.006}$ | $4.384^{\pm.006}$ | $1.645^{\pm.053}$ |
| | T2M-GPT* | $0.362^{\pm.002}$ | $0.507^{\pm.001}$ | $0.593^{\pm.001}$ | $0.335^{\pm.005}$ | $4.247^{\pm.006}$ | $1.984^{\pm.051}$ |
| | MoMask (base) § | $0.375^{\pm.002}$ | $0.534^{\pm.001}$ | $0.628^{\pm.002}$ | $0.557^{\pm.007}$ | $4.022^{\pm.007}$ | $1.654^{\pm.061}$ |
| | MoMask§ | $0.385^{\pm.002}$ | $0.546^{\pm.002}$ | $0.640^{\pm.002}$ | $0.380^{\pm.005}$ | $3.935^{\pm.006}$ | $1.693^{\pm.060}$ |
| | PlanMoGPT (base) | $\underline{0.394}^{\pm.002}$ | $\underline{0.551}^{\pm.001}$ | $\underline{0.641}^{\pm.001}$ | $\underline{0.189}^{\pm.003}$ | $\underline{3.873}^{\pm.003}$ | $\underline{2.461}^{\pm.107}$ |
| | **PlanMoGPT** | $\mathbf{0.401}^{\pm.001}$ | $\mathbf{0.558}^{\pm.002}$ | $\mathbf{0.647}^{\pm.002}$ | $\mathbf{0.141}^{\pm.002}$ | $\mathbf{3.814}^{\pm.012}$ | $\mathbf{2.538}^{\pm.114}$ |
| KIT ML++ | T2M-GPT | $0.289^{\pm.003}$ | $0.438^{\pm.004}$ | $0.529^{\pm.003}$ | $0.516^{\pm.014}$ | $4.669^{\pm.011}$ | $1.813^{\pm.056}$ |
| | MoMask (base) § | $0.296^{\pm.004}$ | $0.452^{\pm.004}$ | $0.550^{\pm.003}$ | $0.508^{\pm.010}$ | $4.454^{\pm.012}$ | $1.594^{\pm.046}$ |
| | MoMask§ | $0.305^{\pm.004}$ | $\mathbf{0.461}^{\pm.004}$ | $\underline{0.555}^{\pm.003}$ | $\underline{0.425}^{\pm.016}$ | $\underline{4.403}^{\pm.013}$ | $1.716^{\pm.054}$ |
| | PlanMoGPT (base) | $\underline{0.305}^{\pm.003}$ | $0.456^{\pm.003}$ | $0.546^{\pm.003}$ | $0.545^{\pm.012}$ | $4.460^{\pm.011}$ | $\underline{2.411}^{\pm.101}$ |
| | **PlanMoGPT** | $\mathbf{0.309}^{\pm.003}$ | $\underline{0.460}^{\pm.003}$ | $\mathbf{0.557}^{\pm.004}$ | $\mathbf{0.230}^{\pm.008}$ | $\mathbf{4.388}^{\pm.009}$ | $\mathbf{2.524}^{\pm.112}$ |

Table 2: Comparing our **PlanMoGPT** with baselines on multiple datasets. **Bold** indicates the best result, and underlined indicates the second best result. § indicates using ground-truth motion length as extra information. **PlanMoGPT (base)** means using VQ-VAE without flow matching. **MoMask (Base)** and **BAMM (Base)** refers to using residual VQ-VAE but without residual Transformer. We implement a variant version of the original T2M-GPT, denoted as **T2M-GPT***, which only differs from PlanMoGPT (base) in that there is no progressive planning. MoMask and T2M-GPT are retrained by their source code on the HumanML3D++ and KIT-ML++ datasets.

training epochs. For efficient flow-matching inference, motion sequences are split into 64-frame clips and re-stitched after inference. The vector field is constructed using a U-Net Ronneberger et al. (2015) with 3 downsampling and 3 upsampling blocks, 256 max channels, and group normalization (32 groups). The HumanML3D and HumanML3D++ datasets share the same motion tokenizer. So do KIT-ML and KIT-ML++. The LLM is implemented by LLaMA-Factory Zheng et al. (2024) by fine-tuning TinyLLaMA (1B parameters) Zhang et al. (2024), with the learning rate on HumanML3D and HumanML3D++ at 5e-5 (1000 warm-up steps), and for KIT-ML and KIT-ML++, it is 1e-3 (600 warm-up steps). Training converges in 25 epochs, significantly fewer than prior works Guo et al. (2024) (500 epochs). Training takes 4.8 hours for VQ-VAE, 15 hours for flow-matching, 28.8 hours for the LLM on HumanML3D, and 149.6 hours on HumanML3D++ on a single NVIDIA H800. KIT-ML and KIT-ML++'s training is faster due to smaller scales.

**Evaluation Metrics.** We follow the metrics from previous works Zhang et al. (2023b); Guo et al. (2024): Frechet Inception Distance (**FID**) score evaluates the quality of the generated motion by measuring the differences in the distribution between the generated and real motions. **R-Precision** and Multimodal Distance (**MM-Dist**) evaluate the semantic alignment between motions and texts. Multimodality (**MModality**) assesses the diversity of motions generated from the same text.

**Baselines.** We compare PlanMoGPT with diffusion methods and token-based methods. For **diffusion-based** baselines, we compare with **MDM** Tevet et al. (2023), **MotionDiffuse** Zhang et al. (2022), **ReMoDiffuse** Zhang et al. (2023c), and **MFM** hu2 (2023); for **token-based** baselines, we compare with **T2M-GPT** Zhang et al. (2023b), **MotionGPT** Jiang et al. (2023), **MotionLLM** Wu et al. (2025), **MotionChain** Jiang et al. (2024), **MoMask** Guo et al. (2024), and **BAMM** Pinyoanuntapong et al. (2024). The MoMask and BAMM without residual learning are also our baselines, denoted as **MoMask (Base)** and **BAMM (Base)**. To validate the effectiveness of the proposed progressive

planning method, we implement a variant of T2M-GPT, referred to as **T2M-GPT\***, as well as TinyLLaMA as the backbone. The only difference between T2M-GPT\* and PlanMoGPT (Base) is that T2M-GPT\* does not integrate the planning mechanism.

## 4.2 Main Results

As shown in Table 2, our PlanMoGPT outperforms existing methods in most metrics on three datasets including HumanML3D++, KIT-ML++, and HumanML3D datasets. Results on various metrics show its effectiveness in semantic alignment and detail preservation.

Our PlanMoGPT achieves substantial improvements in all metrics on the HumanML3D++ dataset, especially an FID (measuring overall quality) of 0.141 (vs. 0.380 in MoMask) and an R@1 score (measuring text-motion alignment) of 40.1% (+1.6% over MoMask). PlanMoGPT scores 2.538 in MModality, also a notable improvement of 0.845 over MoMask. On the KIT-ML++ dataset, our PlanMoGPT also outperforms previous works on most metrics, especially an FID of 0.230 (vs. 0.425 in MoMask) and an MMdoality of 2.524 (vs. 1.716 in MoMask). These all indicate that our method can generate long motions effectively in terms of quality, semantic alignment, and diversity.

Comparing PlanMoGPT to its base version, the proposed flow-enhanced method significantly improves motion quality, particularly in FID, across both datasets. Additionally, PlanMoGPT (Base) surpasses MoMask (Base) and BAMM (Base) in most metrics, and outperforms T2M-GPT\* on both datasets, which validates the effectiveness of our progressive planning method.

PlanMoGPT also achieves new state-of-the-art results on the HumanML3D dataset, with an R@1 score of 52.6%, surpassing BAMM by 0.5%, and an MM-Dist of 2.884, outperforming BAMM by 0.052. For MModality, PlanMoGPT scores 2.971, significantly higher than all prior methods, indicating superior diversity and detail motion generation.

Although our method is effective on multiple datasets, it achieves suboptimal results on the KIT-ML dataset. We attribute this to the smaller scale and lower temporal resolution (12.5fps vs. 20fps for HumanML3D), which may limit the learning of progressive planning. Notably, PlanMoGPT still achieves a high MModality score, which shows that our method can still generate diverse motions on this dataset. More results are provided in the supplementary materials.

## 4.3 Diversity Analysis

We further investigate the relationship between motion diversity and quality. By adjusting the temperature and Top-P parameters during inference, we enable the model to generate motions with varying levels of diversity. The corresponding relationships between these hyperparameters and motion diversity are shown in Section A.7 in the Appendix. Figure 3 compares the performance of the base versions of PlanMoGPT, and non-LLM methods BAMM and Mo-Mask, on the HumanML3D dataset. The performance of BAMM and MoMask significantly drops when the MModality score reaches 2.0 and deteriorates further, or even crashes, when it exceeds 3. This suggests that, while both models perform well at low MModality levels, these non-LLM methods fail to generalize and generate diverse motions due to overfitting. In contrast, PlanMoGPT maintains robust performance across a wide range of MModality levels, showing little decline in FID or R-Precision Top-1

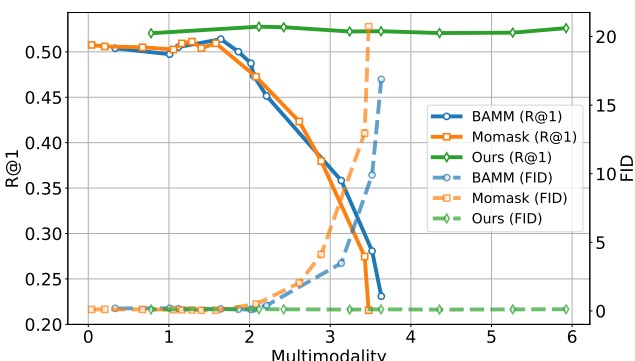

Figure 3: Exploring the relationship between MModality score and quality of generated motions. All these models are the base version. "R@1" refers to R-Precision Top-1.

metrics even when MModality rises to 6. We attribute this advantage to that PlanMoGPT does not have the local dependency issue when generating large interval plans, thus fully exploiting LLMs' generalization power in text-to-motion tasks, which makes the generated plan richer and leads to

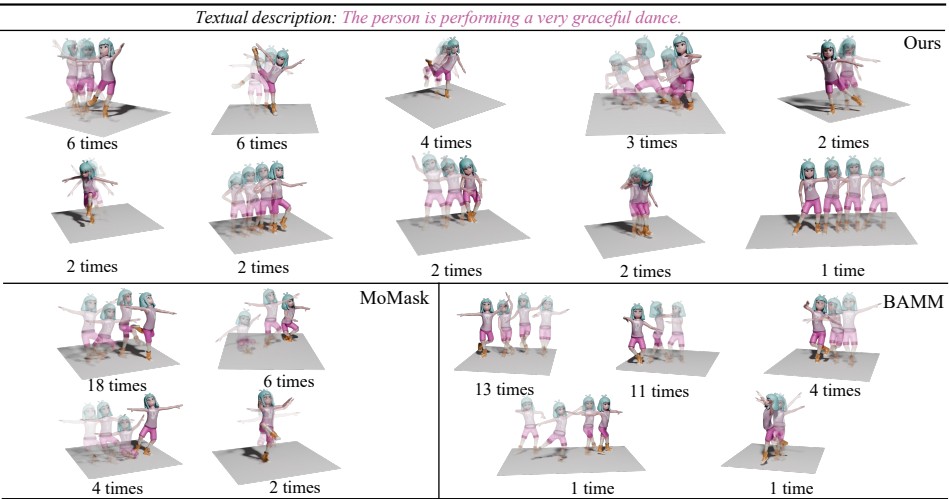

Figure 4: Different methods repeatedly generate 30 motions based on the same text. Similar motions are grouped and reported times.

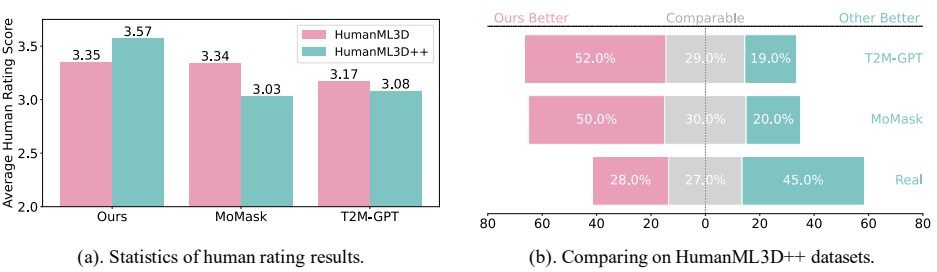

(a). Statistics of human rating results.

(b). Comparing on HumanML3D++ datasets.

Figure 5: Case study on HumanML3D and HumanML3D++ datasets. The score range in Figure (a) ranges from 1 to 5, where 1 means poor and 5 means perfect. Figure (b) compares PlanMoGPT and other methods (ground-truth) to which one generates better results.

diverse motion generation. In Figure 4, we show the results of different methods for repeatedly generating 30 motions based on the same text. It is obvious that the motions generated by our PlanMoGPT are more diverse, i.e., 10 different dancing motions. Although MoMask and BAMM can generate high-quality motions, the motions they generate are relatively similar, i.e., 4 or 5 different dancing motions. This further verifies that our PlanMoGPT can well solve the dilemma of balancing diversity and quality in existing non-LLM research.

## 4.4 SUBJECTIVE EVALUATION

We conduct a subjective evaluation by randomly selecting 200 text samples from the test sets of HumanML3D++ and HumanML3D dataset, and inviting five volunteers to rate the motions generated by different methods. Results are shown in Figure 5. On the HumanML3D and HumanML3D++ datasets, our PlanMoGPT scores 3.35 and 3.57, respectively, significantly better than T2M-GPT and MoMask. Furthermore, the comparison with baselines in the HumanML3D++ dataset in Figure 5 (b) further confirms PlanMoGPT's advantage in generating long-sequence motions.

## 4.5 ABLATION STUDY

**Granularity of Motion Tokenization.** Table 3a shows codebook size and downsampling rate effects. As codebook size increases, both reconstruction and generation performance improve. Notably, with downsampling rate 4, LLM performance improves marginally with larger codebooks, while a downsampling rate of 2 shows substantial gains, surpassing rate 4 at a 4096 codebook size. This indicates that rate 4 requires smaller codebooks as it captures less detail. We therefore select a codebook size of 4096 with a downsampling rate of 2.

| VQ-VAE | Reconstruction | | Generation | |
|---|---|---|---|---|
| | FID ↓ | MPJPE ↓ | FID ↓ | MM-Dist ↓ |
| 512, 4 | $0.088^{\pm.002}$ | 67.6 | $0.185^{\pm.007}$ | $2.987^{\pm.009}$ |
| 512, 2 | $0.146^{\pm.001}$ | 59.0 | $0.244^{\pm.007}$ | $3.045^{\pm.009}$ |
| 1024, 4 | $0.075^{\pm.001}$ | 66.8 | $0.164^{\pm.005}$ | $2.969^{\pm.008}$ |
| 1024, 2 | $0.119^{\pm.000}$ | 59.0 | $0.221^{\pm.007}$ | $2.987^{\pm.008}$ |
| 2048, 4 | $\underline{0.061}^{\pm.001}$ | 66.4 | $0.147^{\pm.004}$ | $2.974^{\pm.007}$ |
| 2048, 2 | $0.074^{\pm.001}$ | **54.4** | $0.142^{\pm.005}$ | $3.002^{\pm.007}$ |
| 4096, 4 | $\mathbf{0.057}^{\pm.000}$ | 66.7 | $\underline{0.130}^{\pm.005}$ | $\underline{2.967}^{\pm.006}$ |
| 4096, 2 | $0.066^{\pm.000}$ | $\underline{56.7}$ | $\mathbf{0.106}^{\pm.004}$ | $\mathbf{2.937}^{\pm.008}$ |

(a) Impact of granularity of motion tokenization.

| VQ-VAE | Size | Model | R@1 ↑ | FID ↓ | MM-Dist ↓ |
|---|---|---|---|---|---|
| Residual | 512, 4 | Base | $0.375^{\pm.002}$ | $0.431^{\pm.005}$ | $4.031^{\pm.004}$ |
| Residual | 1024, 4 | Base | $0.379^{\pm.002}$ | $0.497^{\pm.006}$ | $4.012^{\pm.004}$ |
| Residual | 2048, 2 | Base | $\underline{0.385}^{\pm.002}$ | $\underline{0.334}^{\pm.004}$ | $\underline{3.970}^{\pm.005}$ |
| Residual | 4096, 2 | Base | $0.378^{\pm.002}$ | $0.480^{\pm.006}$ | $4.077^{\pm.006}$ |
| **Flow** | 4096, 2 | Base | $\mathbf{0.394}^{\pm.002}$ | $\mathbf{0.189}^{\pm.003}$ | $\mathbf{3.873}^{\pm.003}$ |
| Residual | 512, 4 | Full | $0.392^{\pm.002}$ | $0.242^{\pm.004}$ | $3.858^{\pm.004}$ |
| Residual | 1024, 4 | Full | $0.391^{\pm.002}$ | $\underline{0.165}^{\pm.004}$ | $\underline{3.849}^{\pm.004}$ |
| Residual | 2048, 2 | Full | $\underline{0.396}^{\pm.002}$ | $0.188^{\pm.003}$ | $3.857^{\pm.005}$ |
| Residual | 4096, 2 | Full | $0.390^{\pm.002}$ | $0.208^{\pm.004}$ | $3.909^{\pm.005}$ |
| **Flow** | 4096, 2 | Full | $\mathbf{0.401}^{\pm.001}$ | $\mathbf{0.141}^{\pm.002}$ | $\mathbf{3.814}^{\pm.012}$ |

(b) Comparison of residual and flow-enhanced.

Table 3: (a). The impact of granularity of motion tokenization to the PlanMoGPT (Base) on the HumanML3D test dataset. "4096, 2" refers to the size of the codebook is 4096, and the downsampling rate is 2. (b). Ablation analysis of flow-enhanced VQ-VAE and residual VQ-VAE on HumanML3D++ dataset. "4096, 2" refers to the size of the codebook and the downsampling rate is 4096 and 2. "base" refers to not using residual Transformer or flow-enhanced method.

**Flow-enhanced VQ-VAE vs. Residual VQ-VAE.** We compare the flow-enhanced VQ-VAE with the previous residual VQ-VAE, which is proposed to reconstruct motion lost due to vector quantization methods in MoMask Guo et al. (2024). Following MoMask's setting, we train a 6-layer residual VQ-VAE, using its first-layer tokens for the PlanMoGPT (Base) and all six layers' tokens for the residual Transformer. As shown in Table 3b, our flow-enhanced VQ-VAE outperforms residual VQ-VAE, regardless of codebook sizes and downsampling rates. Our analysis suggests that the 6-layer residual VQ-VAE will compromise the quality of the base token, thereby weakening the following base generative model. The reconstruction performance of VQVAEs is reported in the supplementary materials.

**Interval of Progressive Planning.** We investigate the impact of the plans, with the results on the HumanML3D test dataset shown in Table 4. Compared to the model without plans, all the models using plans show improvements in both R@1 and MM-Dist metrics. When the interval of the plan is 2, it significantly reduces FID, indicating that the model can generate more refined motions. When the interval of the plan is 4, R@1 improves while MM-Dist decreases, suggesting that the generated motions are more semantically aligned with the text. Combining these two interval plans leads to improvements both in FID and R@1, showing that the enhancements provided by different plans are cumulative. With an interval of 8, although R@1 increases, FID also rises, indicating that a larger interval plan may reduce the fineness of the generated motions. When all

| Interval $T$ | | | R@1 ↑ | FID ↓ | MM-Dist ↓ |
|---|---|---|---|---|---|
| 8 | 4 | 2 | | | |
| | | | $0.494^{\pm.003}$ | $0.130^{\pm.004}$ | $3.109^{\pm.006}$ |
| ✓ | | | $0.506^{\pm.003}$ | $0.162^{\pm.005}$ | $3.013^{\pm.010}$ |
| | ✓ | | $0.514^{\pm.003}$ | $0.128^{\pm.006}$ | $2.969^{\pm.009}$ |
| | | ✓ | $0.508^{\pm.002}$ | $\mathbf{0.104}^{\pm.004}$ | $3.030^{\pm.008}$ |
| ✓ | ✓ | | $\underline{0.518}^{\pm.003}$ | $0.175^{\pm.004}$ | $\mathbf{2.930}^{\pm.007}$ |
| | ✓ | ✓ | $\mathbf{0.521}^{\pm.003}$ | $\underline{0.106}^{\pm.004}$ | $\underline{2.937}^{\pm.008}$ |
| ✓ | ✓ | ✓ | $\underline{0.519}^{\pm.002}$ | $0.119^{\pm.006}$ | $2.973^{\pm.010}$ |

Table 4: Analysis of progressive planning method based on the HumanML3D test dataset.

three intervals are used, there is no further performance improvement. This shows that our selection of intervals 4 and 2 is sufficient.

# 5 CONCLUSION

In this paper, we address what limits the ability of LLM in text-to-motion generation tasks. We identify the granularity of motion tokenization as a critical bottleneck. That is, fine-grained tokens lead to severe local dependencies, while coarse-grained motion tokens lose motion details. To address this issue, we propose PlanMoGPT, an LLM-based framework integrating progressive planning and flow-enhanced fine-grained motion tokenization. Extensive experiments demonstrate that PlanMoGPT not only generates precise and diverse human motions but also outperforms the state-of-the-art method MoMask in both human and automated evaluations on short and long sequence datasets. What's more, PlanMoGPT resolves the diversity-quality dilemma in existing non-LLM approaches. which further verifies the necessity of exploiting the potential of LLM for text-to-motion tasks. In the future, we will explore more flexible planning, such as manually selected keyframes. In addition, we will also explore how to extend PlanMoGPT to generate motion with expressions and hand movements.

# 6 ETHICS STATEMENT

PlanMoGPT's text-to-motion capabilities hold transformative potential for creative industries, enabling rapid prototyping of animations, game character motions, and robotic behavior simulations. By democratizing high-quality motion generation, it could reduce production costs for small studios and educators developing interactive training materials. However, misuse risks include generating deceptive human-like motions for deepfakes or automated disinformation campaigns. The model's reliance on LLMs may inadvertently propagate societal biases present in training data, potentially reinforcing stereotypes in generated motions (e.g., gender-associated gestures). To mitigate harms, we advocate for strict usage guidelines, bias audits, and motion watermarking. Ethically deployed, this technology could enhance assistive devices for mobility-impaired users by translating intent into natural movements, bridging communication gaps between humans and embodied AI systems.

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

## A APPENDIX

### A.1 THE USE OF LLMS

In this work, LLMs were used only for text polishing purposes. All research ideas, methodological innovations, model designs, and experimental analyses were conceived and executed independently by the authors. The authors bear full responsibility for the originality, innovation, and validity of this research.

### A.2 ANALYSIS OF TIME COST

Figure 6a report the time cost of different methods, which is statistics from 100 cases. PlanMoGPT is slower than T2M-GPT but still faster than diffusion-based MotionDiffuse. Despite the speed gap with T2M-GPT, our approach achieves state-of-the-art motion quality (RP@3 0.81 vs 0.68) This shows that our method achieves a quality-speed balance.

### A.3 TIME STEPS OF FLOW MATCHING

Figure 6b reports the impact of the inference steps of the flow matching. As the steps increase, the FID significantly decreases and converges at around 30 steps.

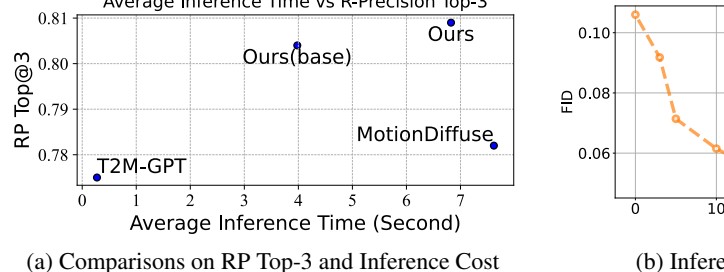
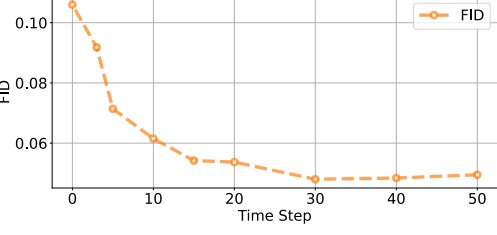

(a) Comparisons on RP Top-3 and Inference Cost          (b) Inference Step of Flow-Matching

Figure 6: (a). Comparisons of average inference time cost. (b). Exploring the inference step of the flow-matching enhanced method on the HumanML3D test dataset.

## A.4 DECODING PROCESS OF MULTI-GRANULARITY MOTION TOKENS

The multi-granularity motion planning strategy employs tokens at different temporal scales to represent hierarchical motion information. Specifically, 4-frame interval tokens capture the global motion structure by summarizing long-span movements and omitting fine details; 2-frame interval tokens provide intermediate-granularity transitions to refine the global structure; and full-sequence tokens encode frame-level details for final motion synthesis. Crucially, only the full-resolution token sequence is directly decoded into the final motion—the coarser-grained tokens serve as intermediate planning layers to guide the structural and temporal coherence of the output. This layered approach enables the model to reason over long-term structure before refining local details, improving both temporal coherence and motion realism. For example, given the prompt "The person is shaking their shoulders, standing up, and then bending at the waist again," the token sequences at different granularities (e.g., [S_4], [S_2], [S_1]) illustrate how higher-level tokens abstract global patterns while finer tokens incrementally add detail.

## A.5 PROTOCOLS OF SUBJECTIVE EVALUATION

The subjective evaluation of generated motions was designed with rigorous quality control measures to ensure reliability despite the limited sample size of five annotators. All annotators underwent standardized training using detailed guidelines covering key motion attributes (e.g., naturalness, fidelity, expressiveness) and were required to pass a qualification test before participation. To mitigate potential bias, 10% of each evaluation batch was independently reviewed by an expert annotator. Batches with inter-rater agreement below 90% relative to expert annotations were discarded and re-annotated. This protocol ensured consistent and high-quality evaluations, reducing subjectivity risks and aligning with best practices for human-centric assessments in motion generation studies.

## A.6 FLOW-ENHANCED TOKENIZER FOR MOMASK FRAMEWORK

The flow-enhanced tokenizer was integrated into the MoMask motion generation framework to evaluate its compatibility and performance. On the HumanML3D test set, the integration yielded consistent improvements across metrics. Specifically, combining the flow-matching method with MoMask's base tokenizer increased the R1 score by +2.8% (from 0.504 to 0.518) and reduced Fréchet Inception Distance (FID) by 39%. Further synergy was observed when flow matching was applied alongside MoMask's residual tokenizer, elevating R1 to 0.528 (a +0.9% gain). The results confirm that the flow-enhanced tokenizer not only maintains compatibility with existing frameworks but also exhibits additive benefits when combined with advanced tokenization strategies, highlighting its robustness and generalizability. The detailed metrics are summarized in the table below.

Table 5: Performance comparison of tokenizer variants on HumanML3D test set.

| Tokenizer | FID | R1 | R2 | R3 | Diversity | Matching |
|---|---|---|---|---|---|---|
| Base | 0.082 | 0.504 | 0.699 | 0.797 | 9.303 | 3.024 |
| Base + Flow Matching | 0.050 | 0.518 | 0.711 | 0.804 | 9.228 | 3.006 |
| Residual | 0.048 | 0.519 | 0.715 | 0.810 | 9.303 | 2.944 |
| Residual + Flow Matching | 0.047 | 0.528 | 0.719 | 0.812 | 9.609 | 2.924 |

## A.7 HYPERPARAMETER SETTINGS FOR DIVERSITY ANALYSIS

We employ the widely adopted Multimodality (MModality) metric from prior works to evaluate motion diversity. For each text prompt, we sample M motions (M=30) under fixed random seeds and compute the MModality score based on these samples.

Temperature and Top-K are two key hyperparameters in the decoding process. The Temperature parameter adjusts the predicted probability distribution as follows:

$$P_{\text{sampled}}(i) = \frac{\exp(p_i/T)}{\sum_j \exp(p_j/T)},$$

where $p_i$ is the predicted logit for the i-th token and T denotes the temperature. Top-P is a decoding parameter that dynamically selects the smallest set of most likely tokens whose cumulative probability

exceeds a predefined threshold (e.g., 0.9), balancing diversity and controllability in text generation. Both parameters significantly affect motion quality (as measured by R@1 and FID) and diversity (as measured by MModality). We include detailed results for PlanMoGPT (with Top-P=0.8), BAMM (Top-P=0.7), and MoMask (Top-P=0.5) under varying temperature settings in Tables 6, respectively.

Table 6: Performance comparison of PlanMoGPT, BAMM, and MoMask under different temperature settings.

| Model | Temperature | MultiModality | R1 | FID |
|---|---|---|---|---|
| PlanMoGPT (Top-P=0.8) | 0.5 | 5.9253 | 0.5261 | 0.1249 |
| | 0.4 | 5.2635 | 0.5211 | 0.1116 |
| | 0.3 | 4.3555 | 0.5207 | 0.0960 |
| | 0.2 | 3.6307 | 0.5227 | 0.1170 |
| | 0.15 | 3.2415 | 0.5224 | 0.0991 |
| | 0.1 | 2.4233 | 0.5271 | 0.1203 |
| | 0.05 | 2.1162 | 0.5277 | 0.1243 |
| | 0.02 | 0.7788 | 0.5205 | 0.1092 |
| BAMM (Top-P=0.7) | 5 | 3.6339 | 0.2309 | 16.8818 |
| | 4 | 3.5198 | 0.2807 | 9.9173 |
| | 3 | 3.1374 | 0.3583 | 3.4761 |
| | 2.0 | 2.2104 | 0.4515 | 0.3954 |
| | 1.6 | 2.0519 | 0.4737 | 0.1409 |
| | 1.2 | 2.0174 | 0.4874 | 0.1087 |
| | 1.0 | 1.8615 | 0.5002 | 0.1220 |
| | 0.8 | 1.6432 | 0.5140 | 0.1478 |
| | 0.4 | 1.1170 | 0.5052 | 0.1657 |
| | 0.1 | 1.0091 | 0.4974 | 0.1939 |
| | 0.02 | 0.3333 | 0.5039 | 0.1852 |
| MoMask (Top-P=0.5) | 20 | 3.4771 | 0.2155 | 20.7158 |
| | 10 | 3.4272 | 0.2746 | 12.9529 |
| | 5 | 2.8887 | 0.3798 | 4.1249 |
| | 4 | 2.6163 | 0.4233 | 2.0409 |
| | 3.0 | 2.0805 | 0.4726 | 0.4917 |
| | 2.0 | 1.5813 | 0.5094 | 0.0536 |
| | 1.6 | 1.4019 | 0.5041 | 0.0505 |
| | 1.2 | 1.2854 | 0.5113 | 0.0680 |
| | 1.0 | 1.1603 | 0.5093 | 0.0784 |
| | 0.8 | 1.0566 | 0.5026 | 0.0935 |
| | 0.4 | 0.6735 | 0.5050 | 0.1109 |
| | 0.1 | 0.2049 | 0.5059 | 0.1152 |
| | 0.02 | 0.0410 | 0.5076 | 0.1128 |

## A.8 TEXT ENCODING IN LLM-BASED MOTION GENERATION

Currently, there exists no unified text encoder architecture for LLM-based or token-based human motion generation methods. Prominent recent approaches, such as MotionGPT-2 and MotionLLM, adopt a decoder-only architecture, directly processing text tokens within a large language model to generate motion tokens. Inspired by this trend towards architectural unification, our study initially explored a similar decoder-only framework. For the purpose of establishing a fair and comparable baseline within this architectural paradigm, we introduced a variant, denoted as "T2M-GPT*", in our experiments. A performance comparison on the HumanML3D dataset across various methods is provided in Table 7.

Table 7: Performance comparison of different text encoder and decoder architectures on the HumanML3D dataset.

| Model | Text Encoder | Decoder | R@1 ↑ | FID ↓ | MM-Dist ↓ |
|---|---|---|---|---|---|
| MotionGPT | T5 (Finetune) | T5 | 0.492 | 0.232 | 3.118 |
| MotionGPT-2 | - | LLaMA-3.1-8B | 0.496 | 0.191 | 3.080 |
| MotionLLM | - | Gemma2-2b-it | 0.515 | 0.230 | 2.967 |
| MoMask | CLIP (Freeze) | Mask Transformer | 0.504 | 0.082 | 3.050 |
| T2M-GPT | CLIP (Freeze) | Causal Transformer | 0.491 | 0.116 | 3.118 |
| T2M-GPT* | - | TinyLLaMA-1.1B | 0.494 | 0.130 | 3.109 |
| PlanMoGPT (Ours) | - | TinyLLaMA-1.1B | **0.521** | **0.106** | **2.937** |

## A.9 Ablation Study on Text Encoder

To thoroughly investigate the impact of the text encoder on model performance and ensure a fair comparison, we conducted an extensive ablation study. This study evaluates our proposed PlanMoGPT and the baseline T2M-GPT under various text encoder configurations on the HumanML3D dataset, with results detailed in Table 8.

Table 8: Ablation study on text encoder and decoder configurations.

| Model | Text Encoder | Decoder | R@1 ↑ | FID ↓ | MM-Dist ↓ |
|---|---|---|---|---|---|
| T2M-GPT (official) | CLIP (Freeze) | Causal Transformer | 0.491 | 0.116 | 3.118 |
| PlanMoGPT | CLIP (Freeze) | Causal Transformer | 0.501 | 0.089 | 3.068 |
| T2M-GPT | CLIP (Finetune) | Causal Transformer | 0.498 | 0.107 | 3.051 |
| PlanMoGPT | CLIP (Finetune) | Causal Transformer | 0.507 | 0.099 | 2.991 |
| T2M-GPT | - | Causal Transformer | 0.381 | 2.106 | 3.829 |
| PlanMoGPT | - | Causal Transformer | 0.422 | 1.683 | 3.326 |
| T2M-GPT | CLIP (Freeze) | TinyLLaMA-1.1B | 0.495 | 0.124 | 3.062 |
| PlanMoGPT | CLIP (Freeze) | TinyLLaMA-1.1B | 0.520 | 0.101 | 2.963 |
| T2M-GPT | CLIP (Finetune) | TinyLLaMA-1.1B | 0.503 | 0.102 | 3.021 |
| PlanMoGPT | CLIP (Finetune) | TinyLLaMA-1.1B | **0.526** | **0.094** | **2.890** |
| T2M-GPT* | - | TinyLLaMA-1.1B | 0.494 | 0.130 | 3.109 |
| PlanMoGPT | - | TinyLLaMA-1.1B | **0.521** | **0.106** | **2.937** |

The experimental results lead to two primary conclusions. First, the performance improvements afforded by our proposed progressive planning method are consistent and effective, irrespective of the specific text encoder configuration. This holds true for models utilizing a frozen pre-trained text encoder, a fine-tuned text encoder, or a purely decoder-only architecture without a separate text encoder. Second, when employing the TinyLLaMA-1.1B model as the decoder, the baseline T2M-GPT architecture shows only marginal performance gains from different text encoders. In contrast, our PlanMoGPT model, which integrates the progressive planning strategy, demonstrates significantly more substantial performance improvements across all evaluated metrics and configurations. Notably, the performance of PlanMoGPT using a frozen CLIP text encoder is highly competitive, nearly matching the results achieved by the decoder-only architecture, which highlights the efficacy of our core method.

## B Evaluation on KIT-ML dataset

The evaluation results on the KIT-ML dataset are shown in Table 9. It achieves suboptimal results on the KIT-ML dataset. We attribute this to the smaller scale and lower temporal resolution (12.5fps vs. 20fps for HumanML3D), which may limit the learning of progressive planning. Notably, PlanMoGPT still achieves a high MModality score, which shows that our method can still generate diverse motions on this dataset. More results are provided in the supplementary materials.

| Datasets | Methods | R-Precision ↑ | | | FID ↓ | MM-Dist ↓ | MModality ↑ |
|---|---|---|---|---|---|---|---|
| | | Top-1 | Top-2 | Top-3 | | | |
| KIT-ML | Real Motion | $0.424^{\pm.005}$ | $0.649^{\pm.006}$ | $0.779^{\pm.006}$ | $0.031^{\pm.004}$ | $2.788^{\pm.012}$ | - |
| | MDM§ | - | - | $0.396^{\pm.004}$ | $0.497^{\pm.021}$ | $1.907^{\pm.214}$ | |
| | MotionDiffuse§ | $0.417^{\pm.004}$ | $0.621^{\pm.004}$ | $0.739^{\pm.004}$ | $1.954^{\pm.062}$ | $2.958^{\pm.005}$ | $0.730^{\pm.013}$ |
| | ReMoDiffuse§ | $0.427^{\pm.014}$ | $0.641^{\pm.004}$ | $0.765^{\pm.055}$ | $\mathbf{0.155}^{\pm.006}$ | $\underline{2.814}^{\pm.012}$ | $1.239^{\pm.028}$ |
| | T2M-GPT | $0.416^{\pm.006}$ | $0.627^{\pm.006}$ | $0.745^{\pm.006}$ | $0.514^{\pm.029}$ | $3.007^{\pm.023}$ | $1.570^{\pm.039}$ |
| | MotionGPT | $0.366^{\pm.005}$ | $0.558^{\pm.004}$ | $0.680^{\pm.005}$ | $0.510^{\pm.016}$ | $3.527^{\pm.021}$ | $2.328^{\pm.117}$ |
| | MoMask (base) § | $0.415^{\pm.010}$ | $0.634^{\pm.011}$ | $0.760^{\pm.005}$ | $0.372^{\pm.020}$ | $2.931^{\pm.041}$ | $1.097^{\pm.054}$ |
| | MoMask§ | $\underline{0.433}^{\pm.007}$ | $\underline{0.656}^{\pm.005}$ | $\underline{0.781}^{\pm.005}$ | $0.204^{\pm.011}$ | $\underline{2.799}^{\pm.022}$ | $1.131^{\pm.043}$ |
| | BAMM§ | $\mathbf{0.436}^{\pm.007}$ | $\mathbf{0.660}^{\pm.006}$ | $\mathbf{0.791}^{\pm.005}$ | $0.200^{\pm.011}$ | $\mathbf{2.714}^{\pm.016}$ | $1.517^{\pm.058}$ |
| | PlanMoGPT (base) | $0.422^{\pm.007}$ | $0.625^{\pm.007}$ | $0.742^{\pm.007}$ | $0.359^{\pm.012}$ | $2.962^{\pm.019}$ | $\mathbf{2.412}^{\pm.131}$ |
| | **PlanMoGPT** | $0.422^{\pm.005}$ | $0.631^{\pm.006}$ | $0.755^{\pm.005}$ | $\underline{0.193}^{\pm.007}$ | $2.964^{\pm.015}$ | $\underline{2.391}^{\pm.109}$ |

Table 9: Comparing our **PlanMoGPT** with baselines on the **KIT-ML** test dataset. **Bold** indicates the best result, and underlined indicates the second best result. § indicates using ground-truth motion length as extra information. **PlanMoGPT (base)** means not including flow-matching refinement. **MoMask (Base)** refers to using residual VQ-VAE but without residual Transformer.

| VQ-VAE | Size | Model | R-Precision ↑ | | | FID ↓ | MM-Dist ↓ | MPJEG ↓ |
|---|---|---|---|---|---|---|---|---|
| | | | Top-1 | Top-2 | Top-3 | | | |
| Residual | 512, 4 | Base | $0.474^{\pm.003}$ | $0.666^{\pm.002}$ | $0.766^{\pm.002}$ | $0.227^{\pm.002}$ | $3.220^{\pm.009}$ | $71.3^{\pm.100}$ |
| Residual | 1024, 4 | Base | $0.485^{\pm.003}$ | $0.676^{\pm.002}$ | $0.773^{\pm.002}$ | $0.172^{\pm.001}$ | $3.162^{\pm.007}$ | $67.0^{\pm.100}$ |
| Residual | 2048, 2 | Base | $0.493^{\pm.002}$ | $0.684^{\pm.003}$ | $0.780^{\pm.002}$ | $0.163^{\pm.000}$ | $3.105^{\pm.007}$ | $64.2^{\pm.100}$ |
| Residual | 4096, 2 | Base | $0.493^{\pm.003}$ | $0.685^{\pm.003}$ | $0.782^{\pm.002}$ | $0.173^{\pm.001}$ | $3.106^{\pm.009}$ | $65.3^{\pm.200}$ |
| Flow | 4096, 2 | Base | $\mathbf{0.502}^{\pm.002}$ | $\mathbf{0.695}^{\pm.003}$ | $\mathbf{0.788}^{\pm.002}$ | $\mathbf{0.066}^{\pm.000}$ | $\mathbf{3.020}^{\pm.006}$ | $\mathbf{56.8}^{\pm.200}$ |
| Residual | 512, 4 | Full | $0.507^{\pm.002}$ | $0.699^{\pm.003}$ | $0.794^{\pm.002}$ | $0.033^{\pm.000}$ | $3.016^{\pm.006}$ | $40.3^{\pm.200}$ |
| Residual | 1024, 4 | Full | $0.506^{\pm.002}$ | $0.698^{\pm.002}$ | $0.794^{\pm.002}$ | $0.022^{\pm.000}$ | $3.019^{\pm.008}$ | $37.4^{\pm.200}$ |
| Residual | 2048, 2 | Full | $0.511^{\pm.003}$ | $0.703^{\pm.002}$ | $\mathbf{0.797}^{\pm.002}$ | $0.022^{\pm.000}$ | $2.996^{\pm.008}$ | $\mathbf{31.6}^{\pm.200}$ |
| Residual | 4096, 2 | Full | $\mathbf{0.512}^{\pm.002}$ | $\mathbf{0.703}^{\pm.003}$ | $0.796^{\pm.002}$ | $0.022^{\pm.000}$ | $\mathbf{2.993}^{\pm.009}$ | $31.7^{\pm.300}$ |
| Flow | 4096, 2 | Full | $0.506^{\pm.003}$ | $0.698^{\pm.002}$ | $0.792^{\pm.002}$ | $\mathbf{0.014}^{\pm.000}$ | $2.995^{\pm.007}$ | $48.1^{\pm.100}$ |

Table 10: Reconstruction of our flow-enhanced VQ-VAE and residual VQ-VAE on the **HumanML3D** test dataset. "base" for "residual" refers to the motion is reconstructed by residual VQ-VAE without residual tokens; "base" for "Flow" refers to our VQ-VAE without flow matching method; "4096, 2" refers to the size of the codebook is 4096, and the downsampling rate is 2.

## C  MOTION RECONSTRUCTION RESULTS

We show the motion reconstruction results of flow-enhanced VQ-VAE and residual VQ-VAE on the HumanML3D dataset in Table 10. Although the 6-layer stacked residual VQ-VAE can achieve better reconstruction results overall, the reconstructed quality from the base token is not suboptimal (corresponding to the "base" in Table 10). This is because residual VQ-VAE disperses the motion details into the residual tokens of each layer, making it difficult for the base tokens to retain detailed information. This will affect the subsequent motion generation model, as shown in Table 5 in the main text, resulting in suboptimal results.

