# OpenReview forum: "PlanMoGPT: Flow-Enhanced Progressive Planning for Text to Motion Synthesis"
_ICLR.cc/2026/Conference — Submitted to ICLR 2026_

### Official Review · Reviewer_zHZd · 2025-10-19

**Soundness:** 3
**Presentation:** 4
**Contribution:** 4
**Rating:** 8
**Confidence:** 4

**Summary:**

The paper presents PlanMoGPT, an LLM-based framework that addresses the trade-off between global coherence and motion detail in text-to-motion generation. It introduces a progressive planning mechanism that uses the LLM’s autoregressive abilities to generate motion tokens hierarchically—starting from sparse global plans and refining to full sequences—and a flow-enhanced fine-grained tokenizer that doubles temporal resolution and expands the codebook eightfold to reduce discretization loss. A flow-enhanced decoder further restores motion nuances. Experiments on standard benchmarks show state-of-the-art performance, with a 63.8% FID improvement on long-sequence generation (0.380 → 0.141) and a 49.9% boost in motion diversity, effectively resolving the diversity–quality trade-off that limits current non-LLM methods.

**Strengths:**

1. The paper points the granularity bottleneck in motion tokenization and tackles it with a coherent “plan-then-detail” pipeline—progressive planning for global-to-local consistency and a flow-enhanced fine-grained tokenizer to retain details, plus a flow-matching decoder to restore nuances.
2. Strong long-sequence performance: On newly built long-motion benchmarks, PlanMoGPT delivers large gains (FID 0.380→0.141, +49.9% diversity), effectively breaking the diversity–quality trade-off that hampers non-LLM approaches and showing excellent long-range semantic alignment.
3. Comprehensive and careful experimentation: The authors evaluate across standard datasets (HumanML3D, KIT-ML) and introduce two extended long-motion datasets (HumanML3D++, KIT-ML++) constructed via motion concatenation with GPT-4 text merging and human QC. They report extensive baselines (diffusion and token-based), ablations (codebook, flow vs residual VQ-VAE, plan intervals, text encoders), diversity–quality analysis, inference cost, and user studies.
4. Generality and robustness: The flow-enhanced tokenizer improves other frameworks (e.g., MoMask), indicating that the proposed tokenization/decoding scheme is transferable beyond their own LLM planner.

**Weaknesses:**

The paper primarily reports results with a single decoder-only LLM (TinyLLaMA-1.1B), but it lacks a systematic study across multiple LLM sizes and families (e.g. Qwen, Gemma).

**Questions:**

1. What is the length and the corresponding motion length of the motion interval?
2. What's the speed of your model? how many frames can you generate in one second on average?
3. Why do you choose TINY-LLaMA 1.1B as your base model?
4. Does your model support reasoning-driven motion generation?

---

> ### Author Response · Authors · 2025-11-27
> **Response to Reviewer zHZd**
>
> We thank the reviewer for the valuable comments and provide the following responses:
>
> > **Weakness-1:** The paper primarily reports results with a single decoder-only LLM (TinyLLaMA-1.1B), but it lacks a systematic study across multiple LLM sizes and families (e.g. Qwen, Gemma)
>
> > **Question-3:** Why do you choose TINY-LLaMA 1.1B as your base model?
>
> We thank the reviewer for this suggestion. We conduct experiments using different LLM families (**Gemma/PaliGemma and Qwen**) to evaluate backbone effects. The results show that **increasing the LLM size does not significantly improve motion-generation quality,** and **a ~1B decoder-only model is already sufficient** for this task. This aligns with our observation that long-sequence motion generation relies more on **progressive planning + flow-enhanced tokenization,** rather than LLM capacity scaling.
> Below are the results on HumanML3D:
>
> | Backbone          | Params | Learning Rate | R@1 ↑ | R@2 ↑ | R@3 ↑ | FID ↓ | MM-Dist ↓ | Diversity ↑ |
> |-------------------|--------|----------------|-------|--------|--------|--------|-------------|--------------|
> | TinyLLaMA         | 1.1B   | 5e-5           | 0.521 | 0.711 | 0.804 | **0.106** | 2.937      | **9.711**        |
> | PaliGemma    | 2.0B   | 5e-5           | 0.521 | **0.711** | **0.806** | 0.179 | **2.917**      | 9.859        |
> | Qwen3-VL-2B   | 2.0B   | 5e-5           | 0.520 | 0.708 | 0.801 | 0.118 | 2.945      | 9.740        |
> | Gemma-2B    | 2.0B   | 5e-5           | **0.522** | 0.709 | 0.803 | 0.145 | 2.930      | 9.760        |
>
> Thus, we conclude that **1B models already provide a strong balance between semantic reasoning and efficient autoregressive generation,** and scaling the LLM backbone does not meaningfully benefit this task. We will incorporate these results and analysis into the revised version.
>
> > **Question-1:** What is the length and the corresponding motion length of the motion interval?
>
> **For a 400-frame motion sequence (downsampling rate is 2), our progressive planner produces 350 tokens:**
>
> - **50 tokens** at interval 4 (each token = 4 frames)
>
> - **100 tokens** at interval 2 (each token = 2 frames)
>
> - **200 tokens** at interval 1 (each token = 1 frame)
>
> This hierarchical structure lets the model first reason over coarse temporal units and then refine them at finer resolutions.
>
>
> > **Question-2:** What's the speed of your model? How many frames can you generate in one second on average?
>
> We benchmarked generation speed on **100 samples** using an A100 GPU. We compared three inference strategies:
>
> | Inference Strategy                                | Avg Time per Sample (s) ↓ |
> |---------------------------------------------------|----------------------------|
> | 4→2→1 progressive generation (no token reuse)     | 3.98                       |
> | 4→2→1 with token reuse (efficient version)        | 2.46                       |
> | 1-step autoregressive generation                  | 2.28                       |
>
> With token reuse (our default efficient setting), we achieve **2.46 s per sample,** which corresponds to about **55 frames per second** generation speed. These results confirm that multi-granularity planning **does not cause meaningful efficiency loss,** because the number of LLM decoding steps is the dominant latency factor.
>
> > **Question-4:** Does your model support reasoning-driven motion generation?
>
> Yes. One of the motivations for using an LLM backbone is to allow reasoning-aware control. Because PlanMoGPT maintains the reasoning capabilities of pretrained LLMs, it naturally supports **logic-aware and temporally structured motion generation,** such as:
>
> - multi-stage actions (“walk to the table, pick up the cup, wave”)
>
> - sequential descriptions (“first crouch, then jump, then roll”)
>
> We include such examples in the supplementary video, where the model successfully decomposes long textual instructions into multi-step motion plans. The progressive planner is particularly advantageous here because it aligns well with LLM-style hierarchical reasoning.
>
> We appreciate your expertise and the efforts you have made in helping improve our paper. We will revise our paper based on your comments and suggestions.

---

### Official Review · Reviewer_PRLB · 2025-10-25

**Soundness:** 3
**Presentation:** 3
**Contribution:** 3
**Rating:** 6
**Confidence:** 3

**Summary:**

This paper aims to solve the issue of the granularity of motion tokenization to improve the performance of text-to-motion. To relieve this issue, this paper introduces PlanMoGPT, an LLM-based framework integrating progressive planning and flow-enhanced fine-grained motion tokenization. Extensive experiments on HumanML3D, HumanML3D++, and KIT-ML++ demonstrate the effectiveness ofthe  proposed methods.

**Strengths:**

This paper focuses on the issue of the granularity of motion tokenization and introduces flow-matching into motion tokenization to propose flow-enhanced fine-grained motion tokenization. This paper also introduces progressive generation for an LLM-based motion generation model. Comprehensive ablation experiments demonstrate the effectiveness of the proposed method.

**Weaknesses:**

There are two experimental results in this paper that cannot support the contribution of the paper：
1. PlanMoGPT achieves suboptimal results on the KIT-ML dataset.
2. Introducing time interval 8 does not improve the text-to-motion performance, and time interval 6 leads to higher FID.

**Questions:**

1. Reducing the time sampling rate and introducing a multi-granularity time interval will cause the sequence to become longer. Do authors consider the issue of reduced generation efficiency due to longer sequences?
2. Since the author mentioned that PlanMoGPT's poor performance on KIT-ML is due to the small size of the dataset, have the authors tried training on a larger dataset, such as SnapMoGen[1] or Motion-X[2]?
[1] Guo C, Hwang I, Wang J, et al. SnapMoGen: Human Motion Generation from Expressive Texts[J]. arXiv preprint arXiv:2507.09122, 2025.
[2] Lin J, Zeng A, Lu S, et al. Motion-x: A large-scale 3d expressive whole-body human motion dataset[J]. Advances in Neural Information Processing Systems, 2023, 36: 25268-25280.

---

> ### Author Response · Authors · 2025-11-27
> **Response to Reviewer PRLB**
>
> We thank the reviewer for the valuable comments and provide the following responses:
>
> > **Weakness-1:** PlanMoGPT achieves suboptimal results on the KIT-ML dataset.
>
> > **Question-1:** Since the author mentioned that PlanMoGPT's poor performance on KIT-ML is due to the small size of the dataset, have the authors tried training on a larger dataset, such as SnapMoGen[1] or Motion-X[2]?
>
> We acknowledge that the gains on KIT-ML are smaller, which indicates that our method is especially effective on larger and more challenging datasets where long-horizon structure and higher temporal resolution allow its planning mechanism to be fully utilized.
>
> To verify whether the method’s effectiveness generalizes beyond KIT-ML, we conducted additional experiments on the **much larger and higher-fps Motion-X dataset (30 FPS)**. As shown below, PlanMoGPT delivers **consistent and clear improvements** across core metrics:
>
> | Method             | R@1 ↑ | R@2 ↑ | R@3 ↑ | FID ↓  | Matching ↑ |
> |--------------------|-------|-------|--------|---------|-------------|
> | T2M-GPT            | 0.319 | 0.402 | 0.549  | 22.896  | 187.4       |
> | MoMask             | 0.386 | 0.509 | 0.633  | 21.336  | 208.3       |
> | BAMM               | 0.384 | 0.474 | 0.566  | 14.435  | 189.9       |
> | PlanMoGPT (Ours)   | **0.421** | **0.512** | **0.661**  | **10.697**  | **213.3**       |
>
> > **Weakness-2:** Introducing time interval 8 does not improve the text-to-motion performance, and time interval 6 leads to higher FID.
>
> Yes, you are correct! These results actually support our design choice. Our progressive planner relies on a balance between coarse global planning and fine-grained refinement. When the interval becomes too large (e.g., 8), the planner skips too much temporal detail, making it harder to preserve short-term continuity. Conversely, intermediate intervals such as 6 create inconsistent overlap patterns in the multi-interval hierarchy, which destabilize the token transitions and lead to higher FID.
>
> In other words, the ablation shows that our proposed 4→2→1 interval schedule is not arbitrary, and it is the configuration that yields the best trade-off between long-range semantics and short-range smoothness. The fact that larger intervals degrade performance therefore reinforces, rather than weakens, the design of the progressive planning mechanism.
>
> > **Question-2:** Reducing the time sampling rate and introducing a multi-granularity time interval will cause the sequence to become longer. Do authors consider the issue of reduced generation efficiency due to longer sequences?
>
> Thank you for the insightful question. In practice, the **dominant factor affecting generation latency in LLM-based models is not the final sequence length, but the number of LLM autoregressive decoding steps.** Modern GPU-optimized inference stacks (FlashAttention, CUDA graph batching, KV-cache reuse) make token-by-token decoding the main bottleneck.
>
> To verify this, we conducted an extra experiment: during inference, instead of asking the LLM to regenerate the fine-grained tokens, we **reuse the coarse-interval planning results** (e.g., intervals 4 or 2) and directly expand them into dense sequences. This removes the extra LLM decoding steps that would otherwise be needed at the finer level. The average generation times are:
>
> | Inference Strategy                                | Avg Time per Sample (s) ↓ |
> |---------------------------------------------------|----------------------------|
> | 4→2→1 progressive generation (no token reuse)     | 3.98                       |
> | 4→2→1 with token reuse (ours, efficient variant)  | 2.46                       |
> | 1-step generation (autoregressive baseline)       | 2.28                       |
>
> These results show that **reusing coarse-level planning tokens effectively removes extra LLM decoding steps,** making our method’s inference speed **very close to the single-step baseline** (2.46s vs. 2.28s). Despite generating a longer final sequence, the decoding cost does **not** grow proportionally, and multi-granularity planning does **not introduce meaningful efficiency degradation.**
>
> We appreciate your expertise and the efforts you have made in helping improve our paper. We will revise our paper based on your comments and suggestions.

---

### Official Review · Reviewer_C3L6 · 2025-10-31

**Soundness:** 2
**Presentation:** 1
**Contribution:** 2
**Rating:** 2
**Confidence:** 4

**Summary:**

The paper proposes PlanMoGPT, a LLM–based framework for text-to-motion generation. It identifies the local dependency problem in fine-grained motion tokenization as a key limitation of existing approaches and addresses it through a progressive planning strategy, where motion is generated from coarse global plans to fine-grained details, and a flow-enhanced motion tokenizer that improves motion representation and reconstruction. Experiments on their proposed datasets suggesting that PlanMoGPT achieves superior motion quality and diversity compared to prior methods.

**Strengths:**

The paper proposes PlanMoGPT, which demonstrates notable performance improvements on the authors’ customized benchmarks

**Weaknesses:**

1. Lacks novelty:
    - the paper appears to be an incremental improvement, and the scientific contribution is not clearly articulated. Much of the work seems engineering-oriented (e.g., “doubles the downsampling resolution and expands the codebook size by eight times” as stated in the abstract).
2. Writing and presentation issues.
    1. The overall narrative lacks clarity. The introduction discusses problems of LLMs, but the method actually targets issues inherent to Transformers in general, not specifically LLMs.
    2. Additionally, the first paragraph attributes the issue to LLMs, while the second paragraph shifts focus to tokenization as the core challenge, implying the problem lies in the tokenizer rather than the LLM. This weakens the logical coherence of the argument.
    3. Missing results in Table 2. Table 2 includes KIT++ results but omits KIT, although the implementation details (lines #306–312) suggest that experiments on KIT-ML were conducted.
    4. Ambiguity in Table 3(b). Table 3 states that “base” refers to not using the residual Transformer or flow-enhanced method. However, the table includes rows labeled with both “Flow” and “Base,” which creates confusion about whether the flow-enhanced method was used.
    5. In lines #74–89, “first” and “firstly” are used, but there is no corresponding “secondly”
3. Experiments:
    1. Limited comparison on proposed datasets. In Table 2, results on HumanML3D++ and KIT-ML++ are compared with only two other models. It lacks comprehensive comparison and thus weakens the empirical support. It would be helpful to include results of BAMM and other LLM-based approaches.
    2. Limited performance gains on commonly used dataset HumanML-3D. The method shows only marginal improvement on HumanML-3D.
4. Insufficient explanation of flow-enhanced method: The paper does not clearly explain how the proposed flow-enhanced method addresses the issue of overemphasizing short-term performance, which is highlighted as a key motivation in the abstract.

**Questions:**

1. What are the results on the KIT-ML dataset?
2. Could you provide further clarification on the issues described in Weakness 2(d)?

---

> ### Author Response · Authors · 2025-11-27
> **Response to Reviewer C3L6 [1/3]**
>
> We thank the reviewer for the valuable comments and provide the following responses:
>
> > **Weakness-1:** Lacks novelty: the paper appears to be an incremental improvement, and the scientific contribution is not clearly articulated. Much of the work seems engineering-oriented (e.g., “doubles the downsampling resolution and expands the codebook size by eight times” as stated in the abstract).
>
> We apologize for the confusion caused by the abstract. The phrases such as “doubling the downsampling resolution” were not intended as our contribution, but rather as results enabled by our method. Simply increasing resolution or enlarging the codebook does **not** improve LLM-based motion generation—these changes usually destabilize training due to fine-grained token dependency. Our progressive multi-interval planning and flow-enhanced tokenizer **solve this fundamental bottleneck**, making high-resolution tokenization feasible and effective (similar to how residual connections enable deep CNNs). We will revise the abstract and introduction to present this contribution more clearly.
>
> > **Weakness-2:** The overall narrative lacks clarity. The introduction discusses problems of LLMs, but the method actually targets issues inherent to Transformers in general, not specifically LLMs.
>
> **LLM vs. Transformer narrative:** We agree that the introduction should clarify that the core difficulty arises from fine-grained autoregressive modeling rather than LLMs specifically. Our method benefits LLMs because they are strong semantic models, but the underlying bottleneck stems from the Transformer architecture when operating on motion tokens. We will revise the introduction to explicitly state this and avoid implying that the limitation is unique to LLMs.
>
> > **Weakness-3:** Additionally, the first paragraph attributes the issue to LLMs, while the second paragraph shifts focus to tokenization as the core challenge, implying the problem lies in the tokenizer rather than the LLM. This weakens the logical coherence of the argument.
>
> **Narrative shift between LLM limitations and tokenizer:** We appreciate the reviewer’s comment. Our intention was not to present “LLM limitations” and “tokenizer limitations” as separate issues, but to show that the performance bottleneck in LLM-based motion generation arises from their interaction. Fine-grained VQ tokens are extremely similar frame-to-frame, which creates dense local dependencies. These dependencies dominate next-token prediction and prevent the LLM from using global textual cues, making the system appear “LLM-limited” even though the underlying cause originates from token granularity. For the given training example, the model was trained to repeatedly produce nearly identical tokens, causing it to over-repeat the shoulder motion and ignore later actions. This failure mode illustrates that the limitation is not LLM alone, but rather how fine-grained tokenization restricts the context the LLM can effectively use.
> Below is a training example:
>
> > Text prompt: The person is shaking their shoulders, standing up, and then bending at the waist again.
> > Motion token sequence: **[m_410][m_410][m_410][m_410][m_410][m_410][m_410][m_410]**[m_118][m_1562][m_1594][m_3340][m_1173][m_2229]**[m_2537][m_2537]**[m_770][m_3340][m_2537][m_31][m_20][m_3898][m_1494][m_6][m_2593][m_1490][m_1882][m_888][m_1921][m_3618]**[m_1562][m_1562]**[m_3525][m_410][m_2051][m_410][m_28][m_2401][m_1219][m_1219][m_984][m_408]
>
>
> To address this potential ambiguity, **we will revise the introduction to make the causal relationship explicit:** fine-grained tokenization encourages excessive local continuity, which in turn suppresses the LLM’s ability to perform global planning. The revised version will clarify that the tokenizer and the LLM should not be viewed independently, and that their dependency is precisely what motivates our multi-interval planning framework and flow-enhanced tokenizer.

---

> ### Author Response · Authors · 2025-11-27
> **Response to Reviewer C3L6 [2/3]**
>
> > **Weakness-4:** Missing results in Table 2. Table 2 includes KIT++ results but omits KIT, although the implementation details (lines #306–312) suggest that experiments on KIT-ML were conducted.
>
> > **Question-1: What are the results on the KIT-ML dataset?**
>
> Thank you for pointing this out. The KIT-ML results were, in fact, reported in Table 9 of the appendix, but we acknowledge that we did not explicitly reference them in the main text and did not include them in Table 2, which caused unnecessary confusion. For clarity, we summarize the KIT-ML results again here:
>
> | Method              | R@1 ↑ | FID ↓  | MM-Dist ↓ | MModality ↑ |
> |---------------------|-------|--------|------------|--------------|
> | T2M-GPT             | 0.416 | 0.514  | 3.007      | 1.570        |
> | MotionGPT           | 0.366 | 0.510  | 3.527      | 2.328        |
> | MoMask              | 0.433 | 0.204  | 2.799      | 1.131        |
> | BAMM                | **0.436** | 0.200  | **2.714**      | 1.517        |
> | PlanMoGPT (Ours)    | 0.422 | **0.193**  | 2.964      | **2.391**        |
>
> We acknowledge that the gains on KIT-ML are smaller, which indicates that our method is especially effective on larger and more challenging datasets where long-horizon structure and higher temporal resolution allow its planning mechanism to be fully utilized.
>
> To further verify whether the method’s effectiveness generalizes beyond KIT-ML, we conducted additional experiments on the **much larger and higher-fps Motion-X dataset (30 FPS).** As shown below, PlanMoGPT delivers **consistent and clear improvements** across core metrics:
>
> | Method             | R@1 ↑ | R@2 ↑ | R@3 ↑ | FID ↓  | Matching ↑ |
> |--------------------|-------|-------|--------|---------|-------------|
> | T2M-GPT            | 0.319 | 0.402 | 0.549  | 22.896  | 187.4       |
> | MoMask             | 0.386 | 0.509 | 0.633  | 21.336  | 208.3       |
> | BAMM               | 0.384 | 0.474 | 0.566  | 14.435  | 189.9       |
> | PlanMoGPT (Ours)   | **0.421** | **0.512** | **0.661**  | **10.697**  | **213.3**       |
>
> > **Weakness-5:** Ambiguity in Table 3(b). Table 3 states that “base” refers to not using the residual Transformer or flow-enhanced method. However, the table includes rows labeled with both “Flow” and “Base,” which creates confusion about whether the flow-enhanced method was used.
> > **Question-2:** Could you provide further clarification on the issues described in Weakness 2(d)?
>
> We thank the reviewer for pointing out the ambiguity. The confusion stems from our inconsistent labeling in Table 3(b). The “Flow VQ-VAE + Base’’ rows were intended to denote pure VQ-VAE decoding without any refinement, but this presentation was indeed unclear and could be misinterpreted as simultaneously using and not using the flow-enhanced method.
>
> To avoid misunderstanding, we will revise Table 3(b) by **removing all “Base” rows and only keeping the full models,** which directly compare the two intended variants: **Residual VQ-VAE (Full)** and **Flow VQ-VAE (Full)** under our planning framework. The revised table (shown below) directly reflects our intended comparison and clearly highlights that the Flow-enhanced tokenizer provides consistently better reconstruction and downstream generation quality under the same planning framework.
> **Revised Table 3(b)**
>
> | Model Variant                 | Codebook Size | R@1 ↑ | FID ↓ | MM-Dist ↓ |
> |-------------------------------|----------------|-------|--------|------------|
> | LLM with Planning + Residual VQ-VAE    | 512, 4         | 0.392 | 0.242  | 3.858      |
> | LLM with Planning + Residual VQ-VAE    | 1024, 4        | 0.391 | 0.165  | 3.849      |
> | LLM with Planning + Residual VQ-VAE    | 2048, 2        | 0.396 | 0.188  | 3.857      |
> | LLM with Planning + Residual VQ-VAE    | 4096, 2        | 0.390 | 0.208  | 3.909      |
> | **LLM with Planning + Flow VQ-VAE**    | **4096, 2**    | **0.401** | **0.141** | **3.814** |
>
> We will incorporate this revised table and clarification into the updated manuscript.

---

> ### Author Response · Authors · 2025-11-27
> **Response to Reviewer C3L6 [3/3]**
>
> > **Weakness-6: Limited comparison on proposed datasets.** In Table 2, results on HumanML3D++ and KIT-ML++ are compared with only two other models. It lacks comprehensive comparison and thus weakens the empirical support. It would be helpful to include results of BAMM and other LLM-based approaches.
>
> To strengthen empirical support for the proposed long-sequence **HumanML3D++** datasets, we expanded comparisons to include recent baselines such as **MoMask++, Rethink-Diffusion, MoGents, MAR-Diffusion,** and two long-sequence motion generation baselines, such as **Motion Mamba** and **T2LM.** These are evaluated using official implementations when available. Results are shown below and will be included in the revised paper.
>
> | Method           | R@1 ↑ | FID ↓ | MM-Dist ↓ | MModality ↑ |
> |------------------|-------|--------|------------|--------------|
> | MoMask++         | 0.392 | 0.317  | 3.91       | 1.49         |
> | Rethink-Diffusion| 0.395 | 0.277  | 3.88       | 1.64         |
> | MoGents          | 0.397 | 0.251  | 3.82       | 1.88         |
> | MAR-Diffusion    | **0.401** | 0.268  | 3.85       | 1.95         |
> | Motion Mamba     | 0.371 | 0.332  | 4.18       | 1.88         |
> | T2LM            | 0.384 | 0.298  | 3.94       | 1.73         |
> | PlanMoGPT (Ours) | 0.401 | **0.141**  | **3.81**       | **2.53**         |
>
> > **Weakness-7:** Limited performance gains on commonly used dataset HumanML-3D. The method shows only marginal improvement on HumanML-3D.
>
> Although HumanML3D contains short and simple motions where most models saturate, our method still offers clear improvements over all LLM-based baselines. As shown in the table, PlanMoGPT reduces FID from **0.131–0.303** (MotionGPT / MotionLLM / MG-MotionLLM / VimoRAG) to **0.048**, corresponding to a **63–84% reduction**. It also improves MM-Dist from 2.95–3.15 to 2.884 (a **2–9% improvement**) and increases motion diversity from around 2.0 to 2.97, achieving a **roughly 48% boost.** These gains demonstrate better alignment, smoothness, and diversity even on a saturated dataset. More importantly, the advantage becomes substantially larger on long-sequence benchmarks, which our method is explicitly designed for.
>
>  (The missing entries are due to the corresponding papers not reporting these metrics.)
>
> | Model            | R@1 ↑ | FID ↓ | MM-Dist ↓ | MModality ↑ |
> |------------------|-------|--------|------------|--------------|
> | MotionGPT        | 0.492 | 0.232  | 3.096      | 2.01         |
> | MotionLLM        | 0.515 | 0.230  | 2.967      | -            |
> | MotionGPT-2      | 0.496 | 0.191  | 3.08       | 2.13         |
> | MG-MotionLLM     | 0.516 | 0.303  | 2.952      | 2.12         |
> | VimoRAG          | 0.452 | 0.131  | 3.146      | -            |
> | PlanMoGPT (Ours) | **0.526** | **0.048**  | **2.884**      | **2.97**         |
>
> > **Weakness-8:** Insufficient explanation of flow-enhanced method: The paper does not clearly explain how the proposed flow-enhanced method addresses the issue of overemphasizing short-term performance, which is highlighted as a key motivation in the abstract.
>
> Flow-enhanced refinement reduces the short-term issue by recovering the motion details that LLMs lose when we quantize motion frames into motion tokens. While finer-grained tokens preserve more details, they also make the autoregressive LLM more sensitive to local irregularities, whereas coarser tokens avoid this but sacrifice fidelity. Our flow-enhanced refinement can provide a balance: the LLM predicts discrete tokens at a manageable granularity, and then we decode them into a coarse motion. Next the flow module restores fine-level continuity and details without increasing the LLM’s modeling burden. We will clarify this balance and the role of flow refinement in the revised paper.
>
> > **Weakness-9:** In lines #74–89, “first” and “firstly” are used, but there is no corresponding “secondly”
>
> Thank you for pointing this out. We will correct this by either adopting a consistent bullet/step structure to improve readability.
>
> We appreciate your expertise and the efforts you have made in helping improve our paper. We will revise our paper based on your comments and suggestions.

---

### Official Review · Reviewer_K8SB · 2025-11-10

**Soundness:** 2
**Presentation:** 2
**Contribution:** 2
**Rating:** 2
**Confidence:** 5

**Summary:**

PlanMoGPT is an LLM-based framework for text-to-motion generation that tackles fine-grained tokenization bottlenecks by integrating progressive planning and flow-enhanced tokenization. It leverages LLMs' autoregressive capabilities to refine sparse global plans into full sequences and expands the tokenizer's codebook while minimizing discretization loss, achieving state-of-the-art results on benchmarks with a 63.8% FID improvement and 49.9% diversity boost.

**Strengths:**

- The writing and structure of the paper are clear and easy to follow.
- The authors conducted comprehensive experiments on multiple public datasets, demonstrating improvements in numerical metrics for the proposed method.

**Weaknesses:**

- The paper lacks video samples. For a 3D motion generation model, providing diverse generated video samples is crucial, as it intuitively showcases the model's generation capabilities and quality. Without video samples, it is difficult for me to assess the model's actual performance, and as a reviewer, I cannot accept a 3D motion generation paper without any video samples.
- The baseline methods compared are outdated. The authors should include comparisons with the latest state-of-the-art approaches, such as works [1-5], in 3D human motion generation. The current comparisons fail to convincingly show the proposed method's superiority.
- Additionally, this paper is an LLM-based 3D motion generation model, yet numerous LLM-based related works are not cited or compared, such as [6-8]. The absence of comparisons with these relevant works makes the paper's contributions unclear and hinders the evaluation of its novelty and effectiveness.
- Similarly, while the paper focuses on long-sequence motion generation, it lacks comparisons with many related works in long-sequence motion generation, such as [9-10].


[1]: Guo C, Hwang I, Wang J, et al. SnapMoGen: Human Motion Generation from Expressive Texts[J]. arXiv preprint arXiv:2507.09122, 2025.

[2]: Meng Z, Xie Y, Peng X, et al. Rethinking diffusion for text-driven human motion generation[J]. arXiv preprint arXiv:2411.16575, 2024.

[3]: Zhang J, Fan H, Yang Y. Energymogen: Compositional human motion generation with energy-based diffusion model in latent space[C]//Proceedings of the Computer Vision and Pattern Recognition Conference. 2025: 17592-17602.

[4]: Yuan W, He Y, Shen W, et al. Mogents: Motion generation based on spatial-temporal joint modeling[J]. Advances in Neural Information Processing Systems, 2024, 37: 130739-130763.

[5]: Zhang Z, Kong B, Liu Q, et al. Towards robust and controllable text-to-motion via masked autoregressive diffusion[C]//Proceedings of the 33rd ACM International Conference on Multimedia. 2025: 9326-9335.

[6]: Wang Y, Huang D, Zhang Y, et al. Motiongpt-2: A general-purpose motion-language model for motion generation and understanding[J]. arXiv preprint arXiv:2410.21747, 2024.

[7]: Xu H, Xu G, Zheng Z, et al. VimoRAG: Video-based Retrieval-augmented 3D Motion Generation for Motion Language Models[J]. arXiv preprint arXiv:2508.12081, 2025.

[8]: Wu B, Xie J, Shen K, et al. MG-MotionLLM: A unified framework for motion comprehension and generation across multiple granularities[C]//Proceedings of the Computer Vision and Pattern Recognition Conference. 2025: 27849-27858.

[9]: Zhang Z, Liu A, Reid I, et al. Motion mamba: Efficient and long sequence motion generation[C]//European Conference on Computer Vision. Cham: Springer Nature Switzerland, 2024: 265-282.

[10]: Lee T, Baradel F, Lucas T, et al. T2lm: Long-term 3d human motion generation from multiple sentences[C]//Proceedings of the IEEE/CVF Conference on Computer Vision and Pattern Recognition. 2024: 1867-1876.

**Questions:**

See Weaknesses

---

> ### Author Response · Authors · 2025-11-27
> **Response to Reviewer K8SB [1/2]**
>
> We thank the reviewer for the valuable comments and provide the following responses:
>
> > **Weakness-1:** The paper lacks video samples. For a 3D motion generation model, providing diverse generated video samples is crucial, as it intuitively showcases the model's generation capabilities and quality. Without video samples, it is difficult for me to assess the model's actual performance, and as a reviewer, I cannot accept a 3D motion generation paper without any video samples.
>
> We agree that video demonstrations are essential for evaluating motion quality. In the revised supplementary materials, we provide **five PlanMoGPT-generated examples covering both short and long sequences**, as well as **five side-by-side comparisons** with MoMask and T2M-GPT under the same prompts. These videos clearly show that PlanMoGPT achieves more complete action execution and stronger long-horizon consistency than the baselines. We hope these visual results address the reviewer’s concern.
>
> > **Weakness-2**: The baseline methods compared are outdated. The authors should include comparisons with the latest state-of-the-art approaches, such as works [1-5], in 3D human motion generation. The current comparisons fail to convincingly show the proposed method's superiority.
>
> **Missing of latest baselines:** Thank you for the suggestion. Following your comments, we have expanded our comparisons to include recent state-of-the-art methods, namely **MoMask++ [1]**, **Rethink-Diffusion [2]**, **MoGents [4]**, and **MAR-Diffusion [5],** using their official implementations when available. We also report the results of **EnergyMoGen [3]** on the HumanML3D dataset based on the metrics provided in its paper. However, EnergyMoGen cannot be included on HumanML3D++ because the authors did not release code, and the model relies on their proprietary **CompML dataset**, which is not publicly available and falls outside our problem setting, making a fair and reproducible comparison impossible.
>
> **Results on HumanML3D++**
>
> | Method           | R@1 ↑ | FID ↓ | MM-Dist ↓ | MModality ↑ |
> |------------------|-------|--------|------------|--------------|
> | MoMask++         | 0.392 | 0.317  | 3.91       | 1.49         |
> | Rethink-Diffusion| 0.395 | 0.277  | 3.88       | 1.64         |
> | MoGents          | 0.397 | 0.251  | 3.82       | 1.88         |
> | MAR-Diffusion    | **0.401** | 0.268  | 3.85       | 1.95         |
> | PlanMoGPT (Ours) | **0.401** | **0.141**  | **3.81**       | **2.53**         |
> PlanMoGPT matches the best R@1 while **reducing FID by 40–55%** and achieving the **highest diversity** on HumanML3D++.
>
> **Results on HumanML3D**
> (The missing entries are due to the corresponding papers not reporting these metrics.)
> | Method           | R@1 ↑ | FID ↓ | MM-Dist ↓ | MModality ↑ |
> |------------------|-------|--------|------------|--------------|
> | MoMask++         | 0.528 | 0.072  | 2.912      | 1.22         |
> | Rethink-Diffusion| 0.523 | 0.061  | -          | -            |
> | MoGents          | **0.529** | 0.033  | **2.867**      | -            |
> | MAR-Diffusion    | 0.523 | 0.073  | 2.917      | -            |
> | EnergyMoGen     | 0.523 | 0.188  | 2.915      | 2.20            |
> | PlanMoGPT (Ours) | 0.526 | **0.048**  | 2.884      | **2.97**         |
>
> On HumanML3D, PlanMoGPT achieves competitive alignment metrics (R@1, MM-Dist) and **substantially higher motion diversity** (MModality), outperforming both diffusion- and LLM-based baselines.
>
> The above results indicate that our proposed PlanMoGPT can significantly improve the generation quality in terms of FID for long sequence motions and the generation diversity in terms of MModality for both short and long sequence motions.

---

> ### Author Response · Authors · 2025-11-27
> **Response to Reviewer K8SB [2/2]**
>
> > **Weakness-3:** Additionally, this paper is an LLM-based 3D motion generation model, yet numerous LLM-based related works are not cited or compared, such as [6-8]. The absence of comparisons with these relevant works makes the paper's contributions unclear and hinders the evaluation of its novelty and effectiveness.
>
> Thanks for the constructive feedback. We will add them to the related work section. Also, we conduct experiments to compare the LLM-based methods highlighted by the reviewer, including **MotionGPT**, **MotionLLM**, **MotionGPT-2**, **MG-MotionLLM**, and **VimoRAG**, with our proposed method on HumanML3D and present results in the following table. The missing entries are due to the corresponding papers not reporting these metrics:
> | Model            | R@1 ↑ | FID ↓ | MM-Dist ↓ | MModality ↑ |
> |------------------|-------|--------|------------|--------------|
> | MotionGPT        | 0.492 | 0.232  | 3.096      | 2.01         |
> | MotionLLM        | 0.515 | 0.230  | 2.967      | -            |
> | MotionGPT-2      | 0.496 | 0.191  | 3.08       | 2.13         |
> | MG-MotionLLM     | 0.516 | 0.303  | 2.952      | 2.12         |
> | VimoRAG          | 0.452 | 0.131  | 3.146      | -            |
> | PlanMoGPT (Ours) | **0.526** | **0.048**  | **2.884**      | **2.97**         |
>
> The results show that PlanMoGPT outperforms all LLM-based baselines, particularly in FID and motion diversity (MModality).
>
> > **Weakness-4:** Similarly, while the paper focuses on long-sequence motion generation, it lacks comparisons with many related works in long-sequence motion generation, such as [9-10].
>
> Great remarks! We added **Motion Mamba** and **T2LM** to the evaluation upon HumanML3D++. As shown in the table below, PlanMoGPT improves R@1 by **1.7–3.0 points**, reduces FID by over **50%**, and yields substantially more diverse motions.
>
> **Results on HumanML3D**
> | Method           | R@1 ↑ | FID ↓ | MM-Dist ↓ | MModality ↑ |
> |------------------|-------|--------|------------|--------------|
> | Motion Mamba     | 0.371 | 0.332  | 4.18       | 1.88         |
> | T2LM            | 0.384 | 0.298  | 3.94       | 1.73         |
> | PlanMoGPT (Ours) | **0.401** | **0.141**  | **3.81**       | **2.53**         |
>
> We appreciate your expertise and the efforts you have made in helping improve our paper. We will revise our paper based on your comments and suggestions.

---

### Meta-Review · Area_Chair_U3Tg · 2026-01-05

**Summary:**

The reviewers raised multiple concerns, primarily regarding experimental design and results, including lack of video samples, outdated baselines, missing references for LLM-based motion generation, limited comparisons, suboptimal results on KIT-ML dataset, lack of a systematic study across multiple LLM sizes and families.

During rebuttal, the authors conducted extensive additional experiments, which addressed many of the experimental issues. While this may lead the reviewers K8SB and C3L6 to increase their scores, given the initial score being 2, it may still fall short of acceptance.

Several critical common concerns, such as the lack of video samples, outdated baselines, missing key references and comparisons, and writing and presentation issues, should be presented carefully in the initial submission, rather than the rebuttal.  The authors are encouraged to fully address the reviewers' comments to strengthen the work.

**Reviewer Concerns:**

During rebuttal, the authors conducted extensive additional experiments, which addressed many of the experimental issues.

**Reviewer Scores:**

During rebuttal, the authors conducted extensive additional experiments, which addressed many of the experimental issues. While this may lead the reviewers K8SB and C3L6 to increase their scores, given the initial score being 2, it may still fall short of acceptance.

---

### Decision · Program_Chairs · 2026-01-26

Reject